# Leveraging Evidence Priors for Robust Prompt Learning under Noisy Supervision in Vision-Language Models

Junnan Zou [1]    Zhu Teng [1]    Wei Zhang [1]    Ming He [2]    Jianping Fan [2]

## Abstract

Prompt learning for vision-language models (VLMs) often suffers from performance degradation when adapting to downstream tasks with noisy labels. Existing methods that rely on filtering or reconstructing supervision can propagate errors, leading to sharp performance drops. We observe that pre-trained embeddings are resilient to label noise, offering stable references despite limited adaptation. Based on this insight, we propose Evidence-Prompt, a framework built on the evidence prior that enhances prompt learning by integrating stable pre-trained knowledge. We treat prompt learning as a Bayesian reasoning task, where credibility is derived from both supervision-agnostic and supervision-conditioned evidence. This framework effectively combines these sources to infer robust training targets under noisy conditions, enabling stable learning even with high noise levels. Extensive experiments on eight benchmarks with both synthetic and real-world noisy labels demonstrate that our method flattens the accuracy–noise curve and consistently outperforms SOTA methods, with notable gains on OxfordPets dataset at a 75% noise rate (+36.6% under Asym and +14.4% under Sym). Additionally, transferability experiments reveal that incorporating our evidence prior into other SOTA methods results in accuracy improvements ranging from 2.6% to 15.66%.

## 1. Introduction

Vision–Language Models (VLMs) (Devlin et al., 2019; Li et al., 2022; Liu et al., 2023; Chi et al., 2025b), exemplified

[1] School of Computer Science and Technology, Beijing Jiaotong University, Beijing, China [2] AI Lab at Lenovo Research, Beijing, China. Correspondence to: Zhu Teng <zteng@bjtu.edu.cn>, Ming He <heming01@foxmail.com>.

*Proceedings of the 43rd International Conference on Machine Learning*, Seoul, South Korea. PMLR 306, 2026. Copyright 2026 by the author(s).

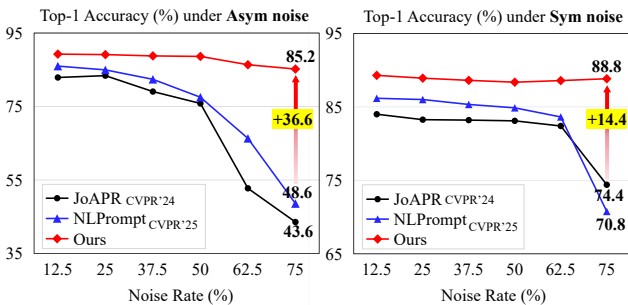

*Figure 1.* Robustness observation of prompt learning methods for VLMs under noisy supervision on OxfordPets. Both Asymmetric (Asym) and Symmetric (Sym) noise settings are considered. Existing methods experience increasingly rapid performance degradation as the noise rate rises. In contrast, our approach consistently maintains robustness, outperforming SOTA methods by **+36.6%** (Asym) and **+14.4%** (Sym) at 75% noise rate.

by CLIP (Radford et al., 2021), learn aligned representations of images and text from large-scale web data, demonstrating remarkable generalization capabilities. Building upon such pre-trained models, prompt learning for VLMs (Zhou et al., 2022; Zhang et al., 2024; Cai et al., 2025; Li et al., 2025b; Zheng et al., 2025) offers a parameter-efficient adaptation strategy for downstream tasks. By freezing the VLM backbone and optimizing only a small set of prompt parameters, it allows for quick transfer to new tasks. However, real-world downstream datasets often contain label noise due to automated collection or annotation errors. Although studies (Wu et al., 2023) suggest that prompt tuning is relatively robust to noisy supervision, excessive noise can still shift prompts toward incorrect semantics, causing cumulative drift and sharp performance drops.

Existing mainstream methods (Wu et al., 2023; Guo & Gu, 2024; Pan et al., 2025) primarily rely on sample splitting or robust losses to address label noise. However, in the context of prompt learning, these methods often suffer from biased early noise estimation, which can lead to incorrect partitions and destabilize training. As shown in Fig. 1, even state-of-the-art prompt learning methods like JoAPR (Guo & Gu, 2024) and NLPrompt (Pan et al., 2025) experience increasingly rapid performance degradation as the noise rate rises. This observation motivates us to ask: can we introduce a reliable reference signal, independent of noisy supervision,

*Table 1.* Performance comparison under symmetric noise (75%) on seven datasets.

| DATASETS | COOP | COOP+PRIOR | OURS |
|---|---|---|---|
| OXFORDPETS | 24.60 | 66.50$^{+41.90}$ | 88.83$^{+64.23}$ |
| CALTECH101 | 46.90 | 76.37$^{+29.47}$ | 90.22$^{+43.32}$ |
| UCF101 | 26.30 | 53.24$^{+26.94}$ | 66.93$^{+40.63}$ |
| FLOWERS102 | 37.17 | 62.97$^{+25.80}$ | 77.71$^{+40.54}$ |
| STANFORDCARS | 22.90 | 38.53$^{+15.63}$ | 61.14$^{+38.24}$ |
| DTD | 17.27 | 28.90$^{+11.63}$ | 50.41$^{+33.14}$ |
| EUROSAT | 26.70 | 35.05$^{+8.35}$ | 56.20$^{+29.50}$ |

to provide stable constraints during optimization?

To investigate this question, we conduct experiments by co-supervising the prompt learning method (e.g., CoOp (Zhou et al., 2022)) with both the training labels and the pre-trained model's probability prior (e.g., CLIP), assigning equal weight to both. The results, as shown in Tab. 1, demonstrate that simply incorporating the pre-trained model's prior significantly alleviates CoOp's degradation under noisy labels. This suggests that, despite its limited adaptation ability, the pre-trained model can serve as a stable reference that is resilient to label noise.

To further rethink these observations, equal-weight supervision is a very coarse heuristic: it ignores sample-wise noise variability and cannot determine when to trust the prior versus the noisy training labels. Moreover, the fixed prior may be misaligned with the target task, offering limited adaptability and restricting further improvements. Based on these insights, we raise a core question: *How can we adaptively leverage the stable prior across varying, unknown noise levels to mitigate performance degradation, while avoiding over-constraining prompt learning so that it can still capture task-specific discrimination?*

From a Bayesian perspective, humans make judgments by integrating cues from multiple sources and updating posterior estimates to reduce uncertainty according to their reliabilities (Knill & Pouget, 2004; Remington et al., 2018). Inspired by this principle, we reformulate noisy supervised learning as a Bayesian posterior reasoning problem: we treat the prior and the noisy supervision signal as two separate sources of observation, estimating their credibility, and perform evidence-conditioned posterior reasoning. Crucially, this process hinges on the precise evaluation of the credibility of both observations, requiring evidence signals that are minimally influenced by noisy labels, yet sufficiently discriminative to maintain task relevance.

Based on the above analysis, we propose a Bayesian reasoning framework for evidence prompt learning in VLMs, termed Evidence-Prompt. First, Evidence-Prompt constructs a sample-wise evidence prior solely from pre-trained representations, eliminating reliance on noisy training labels and providing a relatively reliable reference. Using this evi-

dence prior, we assess the credibility of both distributions from two complementary perspectives: the supervision-agnostic prior is characterized by marginal and joint evidence, while the noisy supervision is evaluated through label-level and semantic evidence. Leveraging these insights, we perform Bayesian posterior reasoning to infer robust targets, thereby stabilizing prompt optimization. As reflected in Fig. 1 and Tab. 1, our method exhibits more stability in performance across all noise ratios and datasets. In summary, our contributions are as follows:

- Noisy labels can degrade model performance in prompt learning for VLMs. Interestingly, we find pre-trained embeddings are resilient to label noise, offering a stable reference despite limited adaptation. Based on this insight, we propose Evidence-Prompt, which is built on the evidence prior to enhance prompt learning under noisy labels. To the best of our knowledge, this is the first work to design an evidence prior in this context.

- We treat prompt learning as a Bayesian reasoning task, where decisions are made by supervision-agnostic credibility and supervision-conditioned credibility. The former is derived from marginal and joint evidence, while the latter is evaluated through label-level and semantic evidence. This novel framework seamlessly integrates these two distributions, significantly enhancing model robustness even under noisy label conditions.

- Extensive experiments across various noisy label settings demonstrate that our method consistently outperforms existing SOTAs, especially on OxfordPets at a 75% noise rate, where it improves +36.6% (Asym) and +14.4% (Sym), respectively, further highlighting its robustness under extreme noise. Additionally, transferability experiments show that incorporating our evidence prior into other SOTA methods can achieve accuracy improvements ranging from 2.6%~15.66%.

## 2. Related Work

### 2.1. Prompt Learning in Vision-Language Models

Prompt learning (PL) offers a parameter-efficient approach to transfer VLMs to downstream tasks (Zheng et al., 2025; Chi et al., 2024; 2025a). Rather than fine-tuning the entire pre-trained encoder, PL optimizes a small set of prompt parameters at the input level, directing the model toward task adaptation with minimal additional cost. Early methods, such as CLIP (Radford et al., 2021), relied on fixed, manually designed templates like "a photo of a [CLASS]". However, Zhou et al. (Zhou et al., 2022) empirically demonstrate that such discrete prompts are highly sensitive to even minor wording changes, making manual trial-and-error unreliable. CoOp (Zhou et al., 2022) replaces discrete templates

with learnable continuous prompt vectors. With frozen image and text encoders, it optimizes prompts end-to-end from a few labeled examples, improving downstream accuracy with high parameter efficiency. ProText (Khattak et al., 2025) distills transferable prompts from LLM-generated contextual text supervision, transferable across classes and datasets. BiomedCoOp (Koleilat et al., 2025) learns contextual prompts with semantic-consistency regularization, statistics-guided prompt selection, and knowledge distillation, improving generalization. While these methods are primarily developed and evaluated under the assumption of clean supervision, focusing on accuracy and transferability, real-world settings often involve noisy annotations. When supervision is corrupted, prompt updates can become misdirected, leading to degraded semantic alignment and reduced generalization stability.

### 2.2. Learning with Noisy Labels

Noisy-label learning seeks robust learning under label noise, reducing the impact of mislabeled supervision. Existing approaches typically enhance robustness by splitting clean/noisy data and performing label correction during training (Li et al., 2024; 2025a; Sheng et al., 2025), or by using noise-insensitive robust objectives (Wang et al., 2025) to stabilize optimization. Recent research has increasingly focused on prompt learning in the presence of label noise. Wu et al. (Wu et al., 2023) show that prompt tuning is comparatively robust to noisy labels, mainly due to the regularization from fixed class tokens and the strong prior in the pretrained vision–language embedding space, and further enhance it with Generalized Cross Entropy (GCE) loss (Zhang & Sabuncu, 2018). To address more challenging noise patterns, subsequent works have introduced partition-based training pipelines. For example, JoAPR (Guo & Gu, 2024) fits a two-component Gaussian Mixture Model (GMM) to the loss distribution in order to separate clean and noisy samples for label refurbishment, while NLPrompt (Pan et al., 2025) investigates the role of Mean Absolute Error (MAE) in prompt learning, uses optimal transport for dataset partitioning, and optimizes clean and noisy subsets with Cross-Entropy and MAE loss (Ghosh et al., 2017), respectively. These methods achieve good performance; however, due to noisy training labels, they are often skewed, leading to significant performance degradation in high-noise settings. To address this issue, we argue that incorporating a noise-agnostic evidence prior is crucial and propose evidence prompt learning to adaptively mitigate noise-induced bias.

## 3. Preliminaries

### 3.1. Problem Definition

Real-world training sets often suffer from noisy labels due to collection and annotation errors, which can disrupt adap-

tation and degrade the semantic alignment and generalization capabilities of VLMs. In this work, we investigate the robustness of these models under unreliable downstream supervision, with a focus on mitigating the negative impact of noisy labels while preserving the inherent strengths of prompt learning. Formally, we are given a noisy training set $D = \{(I_i, \tilde{y}_i)\}_{i=1}^N$, where $I_i$ denotes the $i$-th training image, $\tilde{y}_i \in \{1, \ldots, C\}$ is the observed (potentially corrupted) label, $N$ is the number of training samples, and $C$ is the number of classes. Our goal is to learn stably from corrupted supervision while achieving high recognition accuracy and generalization on test data.

### 3.2. Prompt Learning in Vision-Language Models

Prompt learning has become a key technique for adapting VLMs to downstream tasks. In this work, we focus on enhancing prompt learning using a pre-trained CLIP. To begin, we first review the CLIP framework. CLIP consists of an image encoder $g(\cdot)$ and a text encoder $h(\cdot)$, and performs classification by matching images with corresponding texts. For each class $c$, a class text input $t_c$ is typically constructed using a hand-crafted template (e.g., "a photo of a [CLASS]"). Given an image $I$, CLIP computes the similarity between the image embedding and each class text embedding using Eq. (1), and then derives the class probability distribution through Eq. (2), where $\tau$ is a temperature parameter.

$$s_c = \text{sim}\big(g(I), h(t_c)\big) \tag{1}$$

$$p(y = c \mid I) = \frac{\exp(s_c/\tau)}{\sum_{j=1}^C \exp(s_j/\tau)} \tag{2}$$

The representative prompt-learning method, CoOp, parameterizes the context part of the template with learnable continuous vectors while keeping the image encoder $g(\cdot)$ and text encoder $h(\cdot)$ frozen. Specifically, CoOp introduces a set of learnable context tokens $\{v_1, \ldots, v_m\}$ and concatenates them with the class token to form a new text sequence, as described in Eq. (3). This sequence is fed into the frozen text encoder $h(\cdot)$ to obtain the class text representation. Next, $s_c$ and $p(y = c \mid I)$ are computed using Eq. (1) and Eq. (2). During training, only the prompt parameters are updated using a classification loss, enabling downstream adaptation with minimal trainable parameters.

$$t_c^p = [v_1][v_2] \cdots [v_m][\text{CLASS}] \tag{3}$$

## 4. Method

### 4.1. The Architecture

The proposed Evidence-Prompt addresses noisy-label prompt learning by constructing an evidence prior from pretrained (CLIP) representations and performing credibility-aware Bayesian reasoning for robust optimization. The

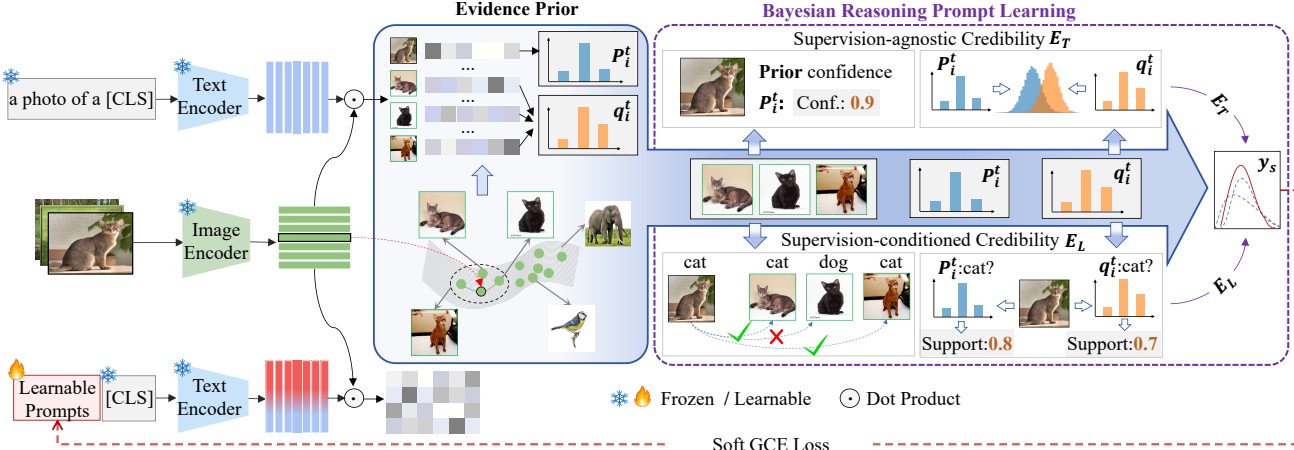

*Figure 2.* The architecture of Evidence-Prompt. It consists of two main components: Evidence Prior and Bayesian Reasoning Prompt Learning. The former provides a noise-agnostic reliable reference; the latter leverages it to estimate credibility and conduct Bayesian posterior reasoning, thereby stabilizing prompt learning.

architecture is illustrated in Fig. 2. For each training sample $I_i$, we first extract the evidence prior $\mathcal{E}_i = \{\mathcal{S}(i), p_i^t, q_i^t\}$. Based on $\mathcal{E}_i$, we estimate sample-wise credibilities: the supervision-agnostic credibility $E_{T,i}$ and the supervision-conditioned credibility $E_{L,i}$. The former assesses the reliability of the supervision-agnostic prior $p_i^t$ by considering both its marginal and joint distribution evidence, while the latter evaluates the reliability of the noisy training label by incorporating label and semantic evidence. Using Bayesian estimation, we infer an evidence-conditioned soft target distribution based on these credibility estimates and optimize the prompts accordingly. This approach stabilizes training and preserves strong performance as label noise increases.

### 4.2. Evidence Prior

To build a reliable evidence prior, we construct a reference signal that does not rely on potentially corrupted training labels. A natural observation is that visual embeddings encoded by pre-trained models like CLIP can form a structured space where visually similar samples are likely to be semantically related. Thus, we extract noise-agnostic evidence directly from these frozen visual representations. Specifically, for each sample $I_i$, we obtain the frozen embedding $z_i = g(I_i)$ and retrieve the top-$K$ most similar samples based on cosine similarity $s_{ij} = z_i^\top z_j$, forming a support pool $\mathcal{S}(i)$ that provides local visual supports for $I_i$. In addition, Eq. (2) gives CLIP semantic probability prior distribution $p_i^t$. However, this single-sample prior only reflects an individual prediction and lacks the corroboration from local visual supports. To address this, we perform a soft voting over the priors within $\mathcal{S}(i)$, yielding a more robust mixture distribution $q_i^t$, as described in Eq. (4). The affinities for this mixture distribution are derived by softmax-normalizing the similarities, as detailed in Eq. (5), where $\sigma_i$

is the standard deviation of similarities within $\mathcal{S}(i)$.

$$q_i^t = \sum_{j \in \mathcal{S}(i)} w_{ij} \, p_j^t \tag{4}$$

$$w_{ij} = \frac{\exp(s_{ij}/(\sigma_i + \epsilon))}{\sum_{k \in \mathcal{S}(i)} \exp(s_{ik}/(\sigma_i + \epsilon))}, \quad j \in \mathcal{S}(i) \tag{5}$$

In summary, we define the sample-wise evidence prior as $\mathcal{E}_i = \{\mathcal{S}(i), p_i^t, q_i^t\}$. Since it is constructed independently of noisy training labels, it can serve as a stable foundation for subsequent credibility estimation.

### 4.3. Bayesian Reasoning for Prompt Learning

In the proposed Evidence-Prompt framework, Bayesian reasoning is crucial for refining prompt learning under noisy labels. We treat the evidence prior $\mathcal{E}_i = \{\mathcal{S}(i), p_i^t, q_i^t\}$ as a key input, guiding the reasoning of the model towards more reliable outputs by assessing the credibility of supervision-agnostic prior and the supervision-conditioned label.

#### 4.3.1. SUPERVISION-AGNOSTIC CREDIBILITY

The reliability of samples should be determined not only by the self-evidence of the prior, but also by corroboration through its local visual supports, which helps prevent high-confidence misclassifications. To this end, we evaluate the supervision-agnostic credibility $E_{T,i}$ from two complementary perspectives: *Marginal Evidence* and *Joint Evidence*.

**Marginal Evidence** quantifies the concentration of the supervision-agnostic prior distribution $p_i^t$ in isolation and is measured by the complement of normalized entropy:

$$E^{\text{me}}(p_i^t) = 1 - \frac{-\sum_{c=1}^{C} p_i^t(c) \log p_i^t(c)}{\log C} \tag{6}$$

**Joint Evidence** evaluates whether the supervision-agnostic prior $p_i^t$ is corroborated by the mixture distribution $q_i^t$. Relying solely on the $p_i^t$ may still be misleading: even a highly concentrated distribution may be unreliable if it contradicts the surrounding mixture distribution. To this end, we incorporate the mixture distribution $q_i^t$ as an external reference. The discrepancy between $p_i^t$ and $q_i^t$ is measured using the Jensen–Shannon divergence as described in Eq. (7).

$$E^{\text{je}}(p_i^t, q_i^t) = \exp\Big(-\text{JS}(p_i^t, q_i^t)\Big) \tag{7}$$

Finally, we synthesize the two perspectives multiplicatively to obtain the evidence prior credibility, as shown in Eq. (8). $E_{T,i}$ achieves a high value only when the evidence prior is both intrinsically certain and strongly corroborated by the mixture distribution. This dual assessment is crucial: reliability hinges not only on the intrinsic certainty of the prior but also on its external validation through the contextual support of the mixture distribution.

$$E_{T,i} = E^{\text{me}}(p_i^t) \cdot E^{\text{je}}(p_i^t, q_i^t) \tag{8}$$

#### 4.3.2. SUPERVISION-CONDITIONED CREDIBILITY

To learn effectively with noisy training labels, it is essential to assess the credibility $E_{L,i}$ of each noisy label $\tilde{y}_i$ accurately, as this plays a key role in enhancing model robustness. We approach this by incorporating two types of evidence: *Noisy Label Evidence* and *Semantic Evidence*. The former leverages the consistency of the support pool when supervision remains reliable. In contrast, the latter provides a noise-agnostic validation of semantic plausibility, helping to mitigate the overconfidence that can arise from noisy labels, especially under conditions of high label noise.

**Noisy Label Evidence.** The construction of Noisy Label Evidence $E_{L,i}^{\text{nl}}$, as defined in Eq. (9), is driven by two key considerations. First, since the noise condition is typically unknown, it is most effective to utilize reliable supervision when it is available for a significant portion of the samples. To this end, we leverage the samples in the support pool that share the same observed label as the current sample to extract label-reliability evidence. Samples with mismatched observed labels ($\tilde{y}_j \neq \tilde{y}_i$) contribute zero to $E_{L,i}^{\text{nl}}$, as specified by Eq. (9), ensuring that only label-matching samples can corroborate $\tilde{y}_i$. Second, even in the presence of noisy annotations, where some samples may coincidentally match $\tilde{y}_j = \tilde{y}_i$, their contributions naturally diminish according to the affinities $w_{ij}$. $E_{L,i}^{\text{nl}}$ increases significantly only when the label is primarily supported by highly similar samples. In contrast, when support mainly comes from weakly similar samples, the accumulated affinities remain small, resulting in a modest $E_{L,i}^{\text{nl}}$.

$$E_{L,i}^{\text{nl}} = \sum_{j \in \mathcal{S}(i)} w_{ij} \, I(\tilde{y}_j = \tilde{y}_i) \tag{9}$$

**Semantic Evidence.** To assess the semantic plausibility of a noisy label and suppress spurious agreements induced by the label, we examine whether $\tilde{y}_i$ receives adequate semantic support under both the supervision-agnostic prior distribution $p_i^t$ and the mixture distribution $q_i^t$. This evidence is denoted as $E_{L,i}^{\text{sem}}$ in Eq. (10).

$$E_{L,i}^{\text{sem}} = \tfrac{1}{2}\big(q_i^t(\tilde{y}_i) + p_i^t(\tilde{y}_i)\big), \tag{10}$$

The final supervision-conditioned credibility $E_{L,i}$ is jointly determined by the noisy label evidence $E_{L,i}^{\text{nl}}$ and the semantic evidence $E_{L,i}^{\text{sem}}$ through the conflict-adaptive mixture in Eq. (11). When $E_{L,i}^{\text{nl}}$ and $E_{L,i}^{\text{sem}}$ diverge significantly, the conflict-adaptive mixture ensures that $E_{L,i}$ is predominantly influenced by the semantic evidence. This mechanism allows $E_{L,i}$ to be more strongly guided by noise-agnostic semantic constraints, thereby mitigating the impact of noise-induced spurious agreement on reliability estimation and optimization.

$$E_{L,i} = \left(1 - \left|E_{L,i}^{\text{nl}} - E_{L,i}^{\text{sem}}\right|\right) E_{L,i}^{\text{nl}} + \left|E_{L,i}^{\text{nl}} - E_{L,i}^{\text{sem}}\right| E_{L,i}^{\text{sem}} \tag{11}$$

#### 4.3.3. BAYESIAN PROMPT LEARNING WITH CREDIBILITY

**Bayesian Posterior Reasoning.** Given the supervision-agnostic credibility $E_{T,i}$ and the supervision-conditioned credibility $E_{L,i}$, our objective is to construct an evidence-conditioned posterior distribution to guide robust Bayesian optimization. In this framework, the supervision-agnostic prior $p_i^t$ represents the prior belief about the true label, while the supervision-conditioned label $\tilde{y}_i$ serves as the observed supervision signal. We construct an evidence-conditioned posterior distribution $y_s$ as a convex combination of the observed supervision $\tilde{y}_i$ and the prior $p_i^t$ distribution, as defined in Eq. (12). Notably, $y_s$ is not a strict realization of Bayes' rule; rather, our method is a Bayesian-inspired, reliability-aware framework with sample-specific adaptive estimation.

$$y_s = (1 - e_i)\,\tilde{y}_i + e_i\,p_i^t \tag{12}$$

$$e_i = \frac{E_{T,i}}{E_{T,i} + E_{L,i} + \epsilon} \tag{13}$$

**Optimization Objective.** We optimize prompt learning using the evidence-conditioned posterior distribution $y_s$, directly aligning the model prediction $p$ with $y_s$. To enhance robustness against noisy supervision, we adopt the generalized cross-entropy (GCE) loss and extend it to distributional targets, as defined in Eq. (14), where $z \in (0, 1]$ is a hyperparameter. The loss is applied to $y_s$, ensuring that the optimization process is jointly guided by both the supervision-agnostic prior and the observed supervision.

$$\mathcal{L}_{\text{SGCE}}(p, y_s) = \frac{1 - \left(\sum_{c=1}^{C} y_s(c)\,p(c)\right)^z}{z} \tag{14}$$

**Behavior Analysis.** This evidence-adaptive posterior is instance-adaptive via $e_i$, leading to two boundary conditions: **1) Supervision-Agnostic Prior Dominated.** When the prior evidence significantly outweighs the observation evidence ($E_{T,i} \gg E_{L,i}$), we have $e_i \to 1$, which results in $y_s \approx p_i^t$. This indicates that the observed label is likely unreliable, and the optimization should be primarily guided by the stable prior, preventing the model from being misled by severely corrupted annotations. **2) Supervision Observation Dominated.** When the observation evidence dominates ($E_{L,i} \gg E_{T,i}$), we have $e_i \to 0$, and thus $y_s \approx \tilde{y}_i$. In this case, the annotation is considered relatively reliable, and the method reduces to near-standard supervised learning, fully leveraging the trustworthy supervision for performance improvement. Overall, by constructing an evidence-driven posterior distribution at the sample level, optimization is guided by reliable signals, making the model less sensitive to noisy annotations, and improving the training stability and robustness of prompt learning.

# 5. Experiments

## 5.1. Experiments Settings

**Datasets.** To evaluate the effectiveness of our method, we conduct experiments on eight datasets, including both synthetically corrupted datasets (OxfordPets (Parkhi et al., 2012), DTD (Cimpoi et al., 2014), Caltech101 (Fei-Fei et al., 2004), UCF101 (Soomro et al., 2012), EuroSAT (Helber et al., 2019), Flowers102 (Nilsback & Zisserman, 2008), StanfordCars (Krause et al., 2013)) and a real-world noisy-label dataset, Food101N (Lee et al., 2018).

**Noise Settings.** For synthetic noisy datasets, we corrupt only the training labels. Following NLPrompt (Pan et al., 2025), we adopt two noise patterns: 1) **Symmetric Noise (Sym):** Each training sample's true label is randomly replaced with any incorrect class with a fixed probability, leading to a nearly uniform noise distribution across classes. 2) **Asymmetric Noise (Asym):** Asymmetric noise introduces class-dependent, directed label flips: samples from the same class are consistently mapped to a neighboring, similar class (the successor class). This corruption pattern is more structured and generally more damaging, making it a stricter benchmark for evaluating noise robustness.

**Implementation Details** We follow the CoOp (Zhou et al., 2022) training protocol and the NLPrompt (Pan et al., 2025) benchmark setup. Models are optimized using SGD with a learning rate of 0.002 and cosine annealing. All methods employ a CLIP (Radford et al., 2021) backbone with a ResNet-50 image encoder (He et al., 2016). We use 16 learnable context tokens shared across classes, with the class token fixed at the end of the prompt. The parameter $z$ in Eq. (14) is set to 0.5 in all experiments, which empirically pro-

vides stable performance across settings. For each dataset, we train on a 16-shot split and report the mean accuracy over three runs with different seeds. All experiments are conducted on a single NVIDIA GeForce RTX 3090 GPU.

## 5.2. Performance Comparison and Analysis

We compare our method with four SOTA methods: CoOp (Zhou et al., 2022), CoOp+GCE (Wu et al., 2023) (denoted as GCE), JoAPR (Guo & Gu, 2024), and NLPrompt (Pan et al., 2025). For the synthetic noisy datasets, as shown in Table 2, we perform a systematic comparison of Evidence-Prompt across 7 benchmarks, under two types of noise (Sym/Asym) and multiple noise ratios. Overall, Evidence-Prompt achieves the best or tied-best performance on all datasets, with increasingly clear advantages in the high-noise regime. Compared to the competitive method, NLPrompt at 75.0% noise, our method achieves 88.83% on OxfordPets under Sym noise, outperforming NLPrompt (70.77%) by 18.06. Under the more challenging, Asym noise, our Evidence-Prompt shows even larger gains, e.g., 85.20% on OxfordPets (+36.60) and 50.71% on DTD (+22.34), demonstrating stable learnability under extreme asymmetric corruption. Table 3 summarizes the test accuracy on the real-world noisy benchmark Food101N. Evidence-Prompt attains 79.24%, outperforming all competing methods and validating its robustness under practical noisy-label supervision.

To investigate robustness, we analyze $\Delta = Acc(12.5\%) - Acc(75.0\%)$, where smaller indicates slower degradation as noise increases. Evidence-Prompt consistently yields the smallest $\Delta$ across all datasets. For instance, on OxfordPets (Sym), Evidence-Prompt achieves an exceptionally low $\Delta$ of 0.46%, compared to the second-best JoAPR with 9.60%. On Caltech101, our method achieves $\Delta$ values of 1.79%/2.35% (Sym/Asym), substantially smaller than competing methods, reflecting more gradual performance decay under severe noise. For Flowers102, whose fine-grained nature and high inter-class similarity make it susceptible to systematic mislabeling, Evidence-Prompt remains stable at high noise (Asym-75) and surpasses NLPrompt by +13.37.

We also report the noisy-to-clean gradient norm ratio during training under 50% noise in Fig. 3. Across datasets, our method consistently maintains a lower and much smoother ratio than NLPrompt, indicating that optimization is less dominated by noise-induced gradients. Notably, NLPrompt exhibits frequent sharp spikes and an overall upward drift, suggesting intermittent noise-driven updates and increasing sensitivity to corrupted labels. In contrast, our Evidence-Prompt suppresses unreliable supervision and anchors learning with noise-agnostic evidence priors, leading to stable gradient dynamics and stronger robustness to label noise.

*Table 2.* Performance evaluations (%) across 7 datasets. $\Delta = Acc(12.5\%) - Acc(75.0\%)$. Best results are in **bold**.

| DATASET | METHOD | NOISE RATE: SYM | | | | | | | NOISE RATE: ASYM | | | | | | |
|---|---|---|---|---|---|---|---|---|---|---|---|---|---|---|---|
| | | 12.5% | 25.0% | 37.5% | 50.0% | 62.5% | 75.0% | Δ(↓) | 12.5% | 25.0% | 37.5% | 50.0% | 62.5% | 75.0% | Δ(↓) |
| OXFORDPETS | CoOp | 76.50 | 66.73 | 60.33 | 47.03 | 35.77 | 24.60 | 51.90 | 76.10 | 66.20 | 52.53 | 38.73 | 26.63 | 14.90 | 61.20 |
| | GCE | 85.63 | 84.60 | 83.67 | 79.23 | 71.40 | 53.17 | 32.46 | 85.50 | 83.03 | 76.73 | 68.07 | 50.70 | 31.97 | 53.53 |
| | JoAPR | 84.00 | 83.26 | 83.20 | 83.10 | 82.40 | 74.40 | 9.60 | 82.90 | 83.40 | 79.07 | 75.84 | 52.74 | 43.57 | 39.33 |
| | NLPrompt | 86.17 | 86.00 | 85.33 | 84.87 | 83.63 | 70.77 | 15.40 | 86.00 | 84.97 | 82.40 | 77.53 | 66.33 | 48.60 | 37.40 |
| | Ours | **89.29** | **88.91** | **88.61** | **88.36** | **88.58** | **88.83** | **0.46** | **89.26** | **89.13** | **88.77** | **88.63** | **86.37** | **85.20** | **4.06** |
| DTD | CoOp | 56.00 | 49.57 | 43.30 | 34.37 | 27.83 | 17.27 | 38.73 | 55.60 | 47.75 | 38.07 | 29.63 | 20.53 | 11.70 | 43.90 |
| | GCE | 61.00 | 59.83 | 56.80 | 50.73 | 43.60 | 33.67 | 27.33 | 60.70 | 57.57 | 52.70 | 43.97 | 33.40 | 18.23 | 42.47 |
| | JoAPR | 58.07 | 57.70 | 56.33 | 53.03 | 48.05 | 29.90 | 28.17 | 52.40 | 56.63 | 53.10 | 48.93 | 40.20 | 28.26 | 24.14 |
| | NLPrompt | 62.97 | 61.23 | 59.17 | 55.17 | 49.03 | 39.80 | 23.17 | 62.30 | 60.60 | 56.47 | 50.80 | 40.27 | 28.37 | 33.93 |
| | Ours | **63.42** | **62.47** | **60.05** | **57.74** | **53.90** | **50.41** | **13.01** | **63.65** | **61.17** | **58.81** | **55.61** | **52.66** | **50.71** | **12.94** |
| CALTECH101 | CoOp | 86.43 | 81.03 | 76.73 | 70.90 | 61.33 | 46.90 | 39.53 | 84.93 | 75.23 | 62.87 | 49.43 | 33.57 | 20.33 | 64.60 |
| | GCE | 92.00 | 90.90 | 90.80 | 89.30 | 86.70 | 79.03 | 12.97 | 91.27 | 91.20 | 89.73 | 85.80 | 78.20 | 62.07 | 29.20 |
| | JoAPR | 90.30 | 90.45 | 89.90 | 88.27 | 86.93 | 83.93 | 6.37 | 90.30 | 89.30 | 88.30 | 88.73 | 85.80 | 81.90 | 8.40 |
| | NLPrompt | 91.73 | 91.13 | 90.77 | 89.93 | 88.30 | 86.70 | 5.03 | 91.60 | 91.17 | 90.20 | 89.27 | 86.17 | 81.07 | 10.53 |
| | Ours | **92.01** | **91.97** | **91.52** | **91.64** | **90.67** | **90.22** | **1.79** | **92.09** | **91.72** | **91.52** | **90.91** | **90.18** | **89.74** | **2.35** |
| UCF101 | CoOp | 69.03 | 63.40 | 58.23 | 49.73 | 40.83 | 26.30 | 42.73 | 67.23 | 58.07 | 46.47 | 34.43 | 23.67 | 13.17 | 54.06 |
| | GCE | 74.00 | 73.63 | 72.57 | 69.37 | 66.00 | 57.07 | 16.93 | 73.90 | 71.87 | 67.97 | 62.23 | 52.50 | 36.37 | 37.53 |
| | JoAPR | 72.83 | 71.17 | 70.37 | 67.63 | 65.30 | 57.67 | 15.16 | 72.07 | 69.80 | 64.10 | 59.17 | 56.07 | 47.46 | 24.61 |
| | NLPrompt | 74.83 | 73.40 | **72.83** | 70.33 | 68.10 | 60.53 | 14.30 | **74.90** | 73.53 | 71.03 | 65.97 | 58.97 | 49.27 | 25.63 |
| | Ours | **74.94** | **73.75** | 72.54 | **70.50** | **68.36** | **66.93** | **8.01** | 74.49 | **73.72** | **71.77** | **68.75** | **65.69** | **62.52** | **11.97** |
| EUROSAT | CoOp | 76.50 | 69.23 | 61.67 | 52.33 | 37.63 | 26.70 | 49.80 | 76.00 | 66.27 | 53.83 | 41.17 | 28.00 | 17.43 | 58.57 |
| | GCE | 82.13 | 78.60 | 74.67 | 63.13 | 49.67 | 31.40 | 50.73 | 78.23 | 72.70 | 63.63 | 45.30 | 22.90 | 12.10 | 66.13 |
| | JoAPR | 75.13 | 61.10 | 60.90 | 63.63 | 38.97 | 27.33 | 47.80 | 69.37 | 67.30 | 59.40 | 47.60 | 33.93 | 17.50 | 51.87 |
| | NLPrompt | **82.53** | 79.53 | **78.13** | 66.70 | 63.53 | 43.80 | 38.73 | 80.13 | 77.13 | **71.43** | 54.30 | **66.33** | 32.73 | 47.40 |
| | Ours | 81.53 | **79.67** | 77.11 | **68.04** | **65.62** | **56.20** | **25.33** | **80.73** | **77.31** | 69.52 | **56.68** | 37.47 | **34.84** | **45.89** |
| FLOWERS102 | CoOp | 88.93 | 83.50 | 77.93 | 70.10 | 55.60 | 37.17 | 51.76 | 86.97 | 74.70 | 60.43 | 42.60 | 26.53 | 12.60 | 74.37 |
| | GCE | 88.80 | 88.33 | 86.73 | 84.07 | 78.37 | 70.37 | 18.43 | 88.40 | 86.37 | 80.33 | 69.93 | 61.50 | 39.23 | 49.17 |
| | JoAPR | 85.57 | 81.23 | 74.60 | 70.23 | 67.90 | 66.93 | 18.64 | 85.17 | 79.63 | 73.97 | 73.83 | 53.37 | 13.27 | 71.90 |
| | NLPrompt | 93.87 | 92.57 | **92.73** | **89.90** | 84.77 | 76.80 | 17.07 | 93.80 | **93.40** | **91.77** | **81.10** | 73.63 | 55.33 | 38.47 |
| | Ours | **94.11** | **92.73** | 92.00 | 88.02 | **84.97** | **77.71** | **16.40** | **94.07** | 91.47 | 88.94 | 79.58 | **74.06** | **68.70** | 25.37 |
| STANFORDCARS | CoOp | 66.20 | 59.70 | 53.40 | 45.90 | 35.67 | 22.90 | 43.30 | 65.77 | 57.13 | 46.23 | 33.73 | 22.37 | 12.80 | 52.97 |
| | GCE | **69.70** | 66.40 | 66.47 | 63.77 | 59.25 | 50.87 | 18.83 | **70.00** | 66.45 | 61.23 | 53.67 | 39.65 | 26.60 | 43.40 |
| | JoAPR | 68.60 | 66.30 | 62.83 | 56.67 | 48.50 | 39.40 | 29.20 | 66.47 | 61.70 | 51.50 | 42.03 | 30.80 | 22.97 | 43.50 |
| | NLPrompt | 69.37 | 68.80 | **67.20** | 65.63 | 62.83 | 58.30 | 11.07 | 69.77 | **67.53** | 64.23 | 59.03 | 50.90 | 39.50 | 30.27 |
| | Ours | 69.30 | 69.16 | 67.19 | 65.87 | 63.39 | **61.14** | **8.16** | 69.83 | 67.32 | **64.51** | **61.90** | **59.20** | **54.86** | **14.97** |

*Table 3.* Test accuracy (%) on Food101N. Best results are in **bold**.

| METHOD | CoOp | GCE | JoAPR | NLPrompt | Ours |
|---|---|---|---|---|---|
| ACCURACY | 69.50 | 71.32 | 72.57 | 76.46 | **79.24** |

## 5.3. Comparisons with Noisy-Label Learning Methods

We further compare our method with classic noisy-label learning methods, including GCE (Zhang & Sabuncu, 2018), JAL-CE (Wang et al., 2025), JAL-FL (Wang et al., 2025), and OGC-CE (Ye et al., 2025). All methods are adapted to the same CoOp framework. As shown in Tables 4 and 5, our method achieves the best performance on OxfordPets, DTD, EuroSAT, and the real-world noisy dataset Food101N. Compared with the second-best method in Table 4, our method improves accuracy by 6.75%, 9.45%, and 11.38% on OxfordPets, DTD, and EuroSAT under 50% Asym noise, respectively. This indicates that our Evidence-Prompt method still maintains an advantage over these methods in prompt learning.

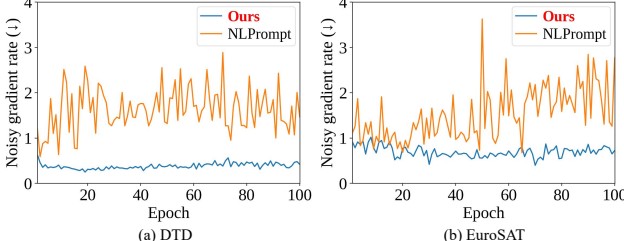

*Figure 3.* Visualizations of noisy-gradient suppression under 50% label noise, measured by the noisy-to-clean gradient norm ratio, which is the ratio of L2 norms on noisy versus clean samples per epoch. Our method consistently achieves a lower ratio than NLPrompt across two datasets, indicating stronger robustness to noise-driven updates.

## 5.4. Transferability of Evidence-Prompt

As shown in Table 6, we evaluate the transferability of Evidence-Prompt by applying it to SOTA methods, JoAPR and NLPrompt, under Sym noise on OxfordPets. Evidence-Prompt consistently yields improvements across all noise

*Table 4.* Comparison with noisy-label learning methods. The best results are highlighted in **bold**.

| DATASET | METHOD | SYM | | ASYM | |
|---|---|---|---|---|---|
| | | 25% | 50% | 25% | 50% |
| OXFORDPETS | GCE | 84.60 | 79.23 | 83.03 | 68.07 |
| | JAL-CE | 83.54 | 79.23 | 83.92 | 60.75 |
| | JAL-FL | 84.30 | 78.80 | 83.65 | 63.37 |
| | OGC-CE | 87.24 | 84.87 | 86.56 | 81.88 |
| | **OURS** | **88.91** | **88.36** | **89.13** | **88.63** |
| DTD | GCE | 59.83 | 50.73 | 57.57 | 43.97 |
| | JAL-CE | 60.70 | 54.20 | 59.75 | 43.20 |
| | JAL-FL | 59.40 | 51.36 | 60.82 | 45.04 |
| | OGC-CE | 58.51 | 54.85 | 59.87 | 46.16 |
| | **OURS** | **62.47** | **57.74** | **61.17** | **55.61** |
| EUROSAT | GCE | 78.60 | 63.13 | 72.70 | 45.30 |
| | JAL-CE | 77.83 | 59.52 | 75.22 | 42.68 |
| | JAL-FL | 77.77 | 61.07 | 73.14 | 41.20 |
| | OGC-CE | 78.64 | 61.07 | 72.35 | 33.10 |
| | **OURS** | **79.67** | **68.04** | **77.31** | **56.68** |

*Table 5.* Comparison with noisy-label learning methods on the real-world noisy dataset Food101N. The best result is in **bold**.

| METHOD | GCE | JAL-CE | JAL-FL | OGC-CE | **OURS** |
|---|---|---|---|---|---|
| ACCURACY | 71.32 | 74.30 | 74.65 | 76.86 | **79.24** |

ratios (12.5%–75.0%), demonstrating that it is not dependent on a specific framework. Notably, at 75.0% noise, it enhances JoAPR and NLPrompt by +11.59 and +15.66, respectively, mitigating severe performance degradation. This suggests that leveraging the evidence prior provides a stable semantic anchor under highly unreliable supervision, reducing noise-induced drift and enhancing robust generalization.

### 5.5. Comparisons based on the Different Backbones

We further evaluate our method with different CLIP backbones, including ViT-B/16 and ViT-L/14, under the symmetric noise setting. As shown in Tables 7 and 8, our method consistently outperforms CoOp and NLPrompt across different datasets, noise levels, and backbone architectures. In particular, under the most challenging 75% symmetric noise setting, our method improves over NLPrompt by 13.90% and 10.39% on OxfordPets with ViT-B/16 and ViT-L/14, respectively. For the real-world noisy dataset Food101N in Table 8, our method also outperforms NLPrompt by 4.16% and 3.91% on the two backbones. These results demonstrate that our evidence-based reliability modeling is not limited to a specific backbone, but can generalize well to stronger vision-language models.

### 5.6. Ablation Study

**Effect of Components:** To investigate the effectiveness of the components in our Evidence-Prompt, we conduct ablation studies on marginal evidence $E_T^{\text{me}}$, joint evidence $E_T^{\text{je}}$,

*Table 6.* Transferability evaluation on **OxfordPets** under the Sym noise setting. The best results are highlighted in **bold**.

| METHOD/NOISE RATIO | 12.5% | 25.0% | 37.5% | 50.0% | 62.5% | 75.0% |
|---|---|---|---|---|---|---|
| JOAPR | 84.00 | 83.26 | 83.20 | 83.10 | 82.40 | 74.40 |
| JOAPR+OURS | **87.22** | **86.62** | **86.67** | **86.45** | **86.48** | **85.99** |
| Δ | +3.22 | +3.36 | +3.47 | +3.35 | +4.08 | +11.59 |
| NLPROMPT | 86.17 | 86.00 | 85.33 | 84.87 | 83.63 | 70.77 |
| NLPROMPT+OURS | **89.56** | **89.26** | **87.93** | **88.25** | **87.65** | **86.43** |
| Δ | +3.39 | +3.26 | +2.60 | +3.38 | +4.02 | +15.66 |

*Table 7.* Performance comparison with different CLIP backbones on OxfordPets and Caltech101 under Sym noise. The best results are highlighted in **bold**.

| BACKBONE | METHOD | 12.5% | 25% | 37.5% | 50% | 62.5% | 75% |
|---|---|---|---|---|---|---|---|
| | | *OxfordPets* | | | | | |
| | CoOp | 86.05 | 76.42 | 67.27 | 60.59 | 42.60 | 30.23 |
| | NLPROMPT | 91.96 | 91.41 | 90.08 | 90.04 | 87.87 | 78.22 |
| | **OURS** | **93.00** | **92.40** | **92.76** | **92.45** | **92.23** | **92.12** |
| ViT-B/16 | | *Caltech101* | | | | | |
| | CoOp | 90.99 | 88.07 | 81.91 | 78.82 | 68.76 | 49.45 |
| | NLPROMPT | 94.97 | 95.54 | 95.17 | 94.04 | 94.00 | 90.83 |
| | **OURS** | **95.78** | **95.86** | **95.90** | **95.94** | **95.90** | **95.46** |
| | | *OxfordPets* | | | | | |
| | CoOp | 84.79 | 78.44 | 69.64 | 61.08 | 47.10 | 30.85 |
| | NLPROMPT | 93.81 | 92.91 | 93.20 | 92.00 | 91.20 | 83.70 |
| | **OURS** | **95.26** | **94.90** | **94.90** | **94.63** | **94.52** | **94.09** |
| ViT-L/14 | | *Caltech101* | | | | | |
| | CoOp | 91.20 | 85.80 | 80.65 | 74.69 | 61.38 | 42.47 |
| | NLPROMPT | 97.16 | 97.36 | 96.84 | 95.17 | 94.81 | 94.46 |
| | **OURS** | **97.77** | **97.65** | **97.73** | **97.48** | **97.61** | **97.20** |

noisy label evidence $E_L^{\text{nl}}$, and semantic evidence $E_L^{\text{sem}}$. The results are presented in Table 9. We begin by examining the naive prior "CoOp + Prior (naive)", which simply combines the pre-trained prior with noisy labels without any evidence modeling. Despite its simplicity, this approach yields substantial gains under heavy noise, improving performance by +8.35% and +4.90% at 75% Sym/Asym noise, respectively. This confirms that the pre-trained prior serves as a stable semantic anchor in the presence of noise. For Evidence-Prompt, the roles of different evidence terms vary with noise severity. 1) Under Sym and Asym-50%, we find that $E_L^{\text{nl}}$ and $E_T^{\text{me}}$ are often the primary sources of performance gains. Removing either one leads to a clear degradation, suggesting that under these noise conditions, corrupted labels typically lack cross-sample consistency and are thus more readily captured and down-weighted by $E_L^{\text{nl}}$; meanwhile, $E_T^{\text{me}}$ characterizes the intrinsic certainty of the supervision-agnostic prior distribution, providing a more reliable reference. 2) Under extreme structured noise (Asym-75), $E_T^{\text{je}}$ and $E_L^{\text{sem}}$ become more critical for suppressing systematic bias induced by severe corruption. For example, removing $E_T^{\text{je}}$ or $E_L^{\text{sem}}$ leads to a sharp drop in performance on Asym-75 (from 34.84 to 10.60 and 11.85), demonstrating that joint and semantic support evidence provide essential constraints. When erroneous supervision forms a seemingly consistent but false pattern, these two types of evidence val-

*Table 8.* Performance comparison with different CLIP backbones on Food101N. The best results are highlighted in **bold**.

| BACKBONE | CoOp | NLPROMPT | OURS |
|---|---|---|---|
| ViT-B/16 | 78.24 | 81.91 | **86.07** |
| ViT-L/14 | 82.68 | 87.40 | **91.31** |

*Table 9.* Ablation studies on Evidence-Prompt components. Performance under Sym and Asym label noise on EuroSAT (%) is reported, with the best results highlighted in **bold**.

| $E_T$ | | $E_L$ | | SYM | | ASYM | |
|---|---|---|---|---|---|---|---|
| $E_T^{\text{me}}$ | $E_T^{\text{je}}$ | $E_L^{\text{nl}}$ | $E_L^{\text{sem}}$ | 50% | 75% | 50% | 75% |
| CoOp | | | | 52.33 | 26.70 | 41.17 | 17.43 |
| CoOp+PRIOR(NAIVE) | | | | 52.47 | 35.05 | 44.68 | 22.33 |
| ✗ | ✓ | ✓ | ✓ | 63.26 | 37.11 | 46.26 | **35.23** |
| ✓ | ✗ | ✓ | ✓ | 65.27 | 50.23 | 53.43 | 10.60 |
| ✓ | ✓ | ✗ | ✓ | 53.81 | 38.17 | 40.68 | 34.69 |
| ✓ | ✓ | ✓ | ✗ | 66.06 | 53.54 | 55.30 | 11.85 |
| ✓ | ✓ | ✓ | ✓ | **68.04** | **56.20** | **56.68** | 34.84 |

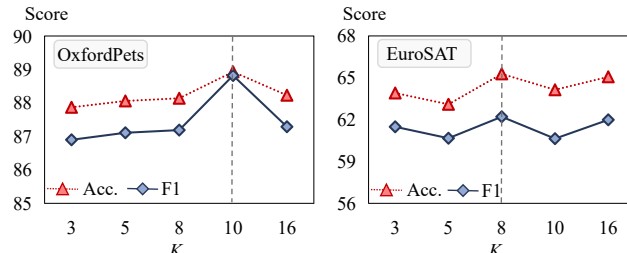

*Figure 4.* Ablation study on support pool size. The accuracy and F1 score curves are reported based on the Top-K threshold division for OxfordPets and EuroSAT with 50% Sym label noise.

idate and down-weight such samples from complementary perspectives—mixture distribution similarity and semantic plausibility—thereby suppressing high-risk updates and preventing performance collapse.

**Effect of Support Pool Size:** To evaluate the impact of support pool size K, we conduct an ablation study on validation set under the Sym-50% setting, using accuracy and F1 score on OxfordPets and EuroSAT, as shown in Fig. 4. Performance on both datasets varies smoothly with K, indicating that our method is relatively insensitive to K and remains robust. Specifically, OxfordPets achieves the best or near-best results around K=10, with diminishing returns as K increases further. EuroSAT peaks at K=8 and remains comparable for larger K. These results suggest that our method maintains stability as K varies, likely because we employ a similarity-driven soft aggregation when constructing evidence. Consequently, newly added support pool samples with low similarity or inconsistent labels are naturally down-weighted, contributing minimally to the final targets and preventing drastic performance fluctuations with K.

## 6. Conclusion

In this work, we investigate the performance degradation of prompt-learning for VLMs during downstream adaptation under noisy annotations. Unlike previous approaches that focus on correcting noisy labels, we observe that the pre-trained model's embedding is largely unaffected by label noise and, despite its limited adaptability, can serve as a stable reference throughout training. Building on this insight, we propose Evidence-Prompt, a Bayesian reasoning-based framework that establishes an evidence prior to characterize the instance-wise credibilities of two supervision sources: the supervision-agnostic pre-trained prior and the supervision-conditioned labels. Extensive experiments demonstrate that our framework achieves robust performance across multiple benchmarks and diverse noise settings. These results emphasize that pre-trained knowledge is not only the starting point for learning but also a persistent anchor for stable optimization under real-world uncertainty. Our Bayesian evidence framework offers a principled way to formalize this anchor, opening new avenues for building reliable and adaptable vision–language systems under imperfect supervision.

## Acknowledgements

This work is sponsored by CCF-Lenovo Blue Ocean Research Fund (CCF-Lenovo OF ST202604), Natural Science Foundation of China (Grant NO. 62472025), and Beijing-Tianjin-Hebei Natural Science Foundation Cooperation Project under grant 25JJJJC0045.

## Impact Statement

The proposed Evidence-Prompt is an evidence-prior-based Bayesian framework that improves the robustness of prompt learning in vision–language models (VLMs) under noisy supervision, aiming to mitigate noise-induced bias and stabilize optimization. This paper presents work whose goal is to advance the field of Machine Learning. There are many potential societal consequences of our work, none which we feel must be specifically highlighted here.

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

# APPENDIX

This supplementary material includes the following:

- Section A summarizes comprehensive descriptions of the datasets used in our experiments.

- Section B presents experiments on the large-scale SUN397 dataset.

- Section C outlines the confusion matrix on OxfordPets dataset.

- Section D provides additional ablation studies to further analyze the roles of different evidence terms.

- Section E provides additional analysis on prior utilization under noisy supervision.

- Section F compares our method with DHO, a distillation-based method that leverages priors, under noisy-label settings.

## A. Details of Dataset Setup

Our experimental benchmark consists of 9 visual classification datasets, where we manually inject noise into the training labels to construct synthetic noisy settings. We further include the real-world noisy dataset Food101N, which naturally contains noisy annotations and thus requires no additional modification. The details are shown in Table 10.

*Table 10.* Datasets details.

| NOISE TYPE | DATASET | TASK | CLASSES | TRAINING SIZE | TESTING SIZE |
|---|---|---|---|---|---|
| | CALTECH101 | OBJECT RECOGNITION | 100 | 4,128 | 2,465 |
| | FLOWERS102 | FINE-GRAINED FLOWERS RECOGNITION | 102 | 4,093 | 2,463 |
| | OXFORDPETS | FINE-GRAINED PETS RECOGNITION | 37 | 2,944 | 3,669 |
| | UCF101 | VIDEO ACTION RECOGNITION | 101 | 7,639 | 3,783 |
| SYNTHETIC NOISY DATASET | DTD | TEXTURE RECOGNITION | 47 | 2,820 | 1,692 |
| | EUROSAT | SATELLITE IMAGE CLASSIFICATION | 10 | 13,500 | 8,100 |
| | STANFORDCARS | FINE-GRAINED CAR RECOGNITION | 196 | 6,509 | 8,041 |
| | SUN397 | SCENE RECOGNITION | 397 | 15,880 | 19,850 |
| REAL-WORLD NOISY DATASET | FOOD101N | FINE-GRAINED FOOD RECOGNITION | 101 | 310,009 | 30,300 |

## B. Experiments on SUN397

We further evaluate our method on SUN397, a large-scale scene dataset with many categories, and report the results in Table 11. Ours consistently achieves the best accuracy under both Sym and Asym noise across all noise rates. More importantly, it exhibits a markedly smaller performance drop ($\Delta$) under both Sym and Asym noise, indicating more stable performance as noise increases. This confirms that our approach maintains more stable performance, especially at high-noise regimes.

*Table 11.* Test accuracy (%) on SUN397. $\Delta = Acc(12.5\%) - Acc(75.0\%)$. Best results are in **bold**.

| DATASET | METHOD | NOISE RATE: SYM | | | | | | | NOISE RATE: ASYM | | | | | | |
|---|---|---|---|---|---|---|---|---|---|---|---|---|---|---|---|
| | | 12.5% | 25% | 37.5% | 50% | 62.5% | 75% | $\Delta$ ($\downarrow$) | 12.5% | 25% | 37.5% | 50% | 62.5% | 75% | $\Delta$ ($\downarrow$) |
| SUN397 | COOP | 65.5 | 62.9 | 59.3 | 55.5 | 48.3 | 37.8 | 27.7 | 63.5 | 56.1 | 45.5 | 33.8 | 22.1 | 11.4 | 52.1 |
| | GCE | 67.6 | 66.3 | 65.4 | 64.2 | 62.0 | 59.2 | 8.4 | 68.4 | 66.4 | 63.8 | 60.0 | 53.6 | 43.8 | 24.6 |
| | NLPROMPT | 68.4 | 67.5 | 66.4 | 64.8 | 64.1 | 61.7 | 6.7 | 68.7 | 67.5 | 66.1 | 64.0 | 61.4 | 53.0 | 15.7 |
| | OURS | **69.4** | **68.2** | **68.4** | **67.7** | **66.9** | **66.2** | **3.2** | **69.3** | **69.1** | **67.6** | **66.9** | **64.7** | **63.0** | **6.3** |

## C. Confusion Matrix

From the confusion matrices in Fig. 5, we can clearly observe that under 75% extremely heavy noise, our method exhibits a sharper and more continuous diagonal on OxfordPets across Sym/Asym settings, with less off-diagonal "leakage", indicating

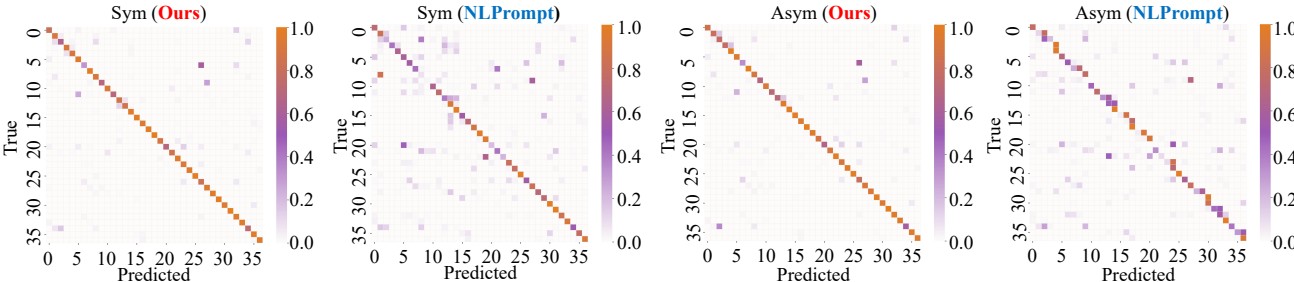

*Figure 5.* Confusion matrix on OxfordPets dataset with 75% noise rate.

a more stable class decision boundary at test time. In contrast, NLPrompt shows a weaker diagonal and more pronounced off-diagonal bright spots/blocks, and is especially prone to structured, systematic confusions under Asym noise. These observations corroborate that our Evidence-Prompt provides a stable reference under both Sym and Asym label noise, leading to more robust generalization.

## D. Additional Ablation Study

Under Sym and moderately structured noise (Asym-50), the performance gains are mainly driven by evidence that can capture the inconsistency of random corruption ($E_T^{\text{me}}$ and $E_L^{\text{nl}}$): keeping only these terms still yields competitive results, indicating that corrupted labels in these regimes typically lack stable cross-sample support and are thus easier to identify and down-weight. In contrast, under extreme structured noise (Asym-75), the constraint effects of $E_T^{\text{je}}$ and $E_L^{\text{sem}}$ become indispensable—removing either one can trigger a cliff-like collapse (Asym-75 drops to around 10% once the semantic constraint is removed), suggesting that when erroneous supervision forms seemingly consistent spurious patterns, joint evidence and semantic feasibility act as complementary constraints from the perspectives of mixture distribution and semantic plausibility, jointly suppressing high-risk updates and preventing collapse. Overall, enabling all evidence terms consistently achieves the strongest performance across noise settings, highlighting the synergy between "gain-driving" evidence and "bias-suppressing" constraints.

*Table 12.* Ablation studies under Sym and Asym noise on EuroSAT (%). Best results are in **bold**.

| $E_T$ | | $E_L$ | | SYM | | ASYM | |
|---|---|---|---|---|---|---|---|
| $E_T^{\text{me}}$ | $E_T^{\text{je}}$ | $E_L^{\text{nl}}$ | $E_L^{\text{sem}}$ | 50% | 75% | 50% | 75% |
| ✓ | ✗ | ✗ | ✓ | 54.73 | 37.81 | 47.10 | 33.73 |
| ✓ | ✗ | ✓ | ✗ | 65.75 | 51.33 | 55.19 | 10.11 |
| ✗ | ✓ | ✗ | ✓ | 50.85 | 37.62 | 39.38 | 34.52 |
| ✗ | ✓ | ✓ | ✗ | 64.40 | 37.95 | 47.62 | 33.56 |
| ✓ | ✓ | ✓ | ✓ | **68.04** | **56.20** | **56.68** | **34.84** |

## E. Additional Analysis on Prior Utilization under Noisy Supervision

We further analyze the role of pretrained CLIP priors under noisy supervision. As shown in Figure 6, zero-shot CLIP remains stable across noise ratios but lacks target-domain adaptation, while NLPrompt suffers clear degradation as label noise increases. In contrast, Evidence-Prompt maintains a flatter accuracy–noise curve under both symmetric and asymmetric noise, indicating better robustness to corrupted supervision while preserving the adaptability of prompt learning.

Tables 13 and 14 further compare CoOp and NLPrompt with their "Original", "+Naive Prior", and "+Ours" variants. Here, "Original" denotes the prompt learner itself without using the CLIP prior during training. Naive prior integration clearly benefits CoOp and improves NLPrompt mainly under severe noise, but it can also degrade NLPrompt under mild noise. In contrast, their "+Ours" variants estimate sample-wise credibility through our Evidence-Prompt strategy, enabling stable performance improvements across different noise levels.

*Table 13.* Comparisons of naive prior integration and our evidence-based integration on **DTD and OxfordPets** under Sym noise. Best results are highlighted in **bold**.

| DATASET | METHOD | VARIANT | SYM | | | | | |
| | | | 12.5% | 25% | 37.5% | 50% | 62.5% | 75% |
|---------|--------|---------|-------|-----|-------|-----|-------|-----|
| DTD | CoOp | ORIGINAL | 56.00 | 49.57 | 43.30 | 34.37 | 27.83 | 17.27 |
| | | +NAIVE PRIOR | 62.23 | 58.51 | 53.07 | 46.99 | 40.13 | 28.90 |
| | | +OURS | **63.42** | **62.47** | **60.05** | **57.74** | **53.90** | **50.41** |
| | NLPROMPT | ORIGINAL | 62.97 | 61.23 | 59.17 | 55.17 | 49.03 | 39.80 |
| | | +NAIVE PRIOR | 60.58 | 60.64 | 58.04 | 55.57 | 50.18 | 43.79 |
| | | +OURS | **63.12** | **63.06** | **60.82** | **58.45** | **55.85** | **51.71** |
| OXFORDPETS | CoOp | ORIGINAL | 76.50 | 66.73 | 60.33 | 47.03 | 35.77 | 24.60 |
| | | +NAIVE PRIOR | 86.16 | 83.67 | 80.89 | 76.12 | 69.12 | 66.50 |
| | | +OURS | **89.29** | **88.91** | **88.61** | **88.36** | **88.58** | **88.83** |
| | NLPROMPT | ORIGINAL | 86.17 | 86.00 | 85.33 | 84.87 | 83.63 | 70.77 |
| | | +NAIVE PRIOR | 85.17 | 85.39 | 84.66 | 83.05 | 81.88 | 78.20 |
| | | +OURS | **89.56** | **89.26** | **87.93** | **88.25** | **87.65** | **86.43** |

*Table 14.* Comparisons of naive prior integration and our evidence-based integration on **Food101N**. Best results are highlighted in **bold**.

| CoOp | ORIGINAL | +NAIVE PRIOR | +OURS |
|------|----------|--------------|-------|
| | 69.50 | 74.40 | **79.24** |

| NLPROMPT | ORIGINAL | +NAIVE PRIOR | +OURS |
|----------|----------|--------------|-------|
| | 76.46 | 77.90 | **78.30** |

# F. Comparison with DHO under Noisy Supervision

We further compare Evidence-Prompt with DHO (Kang et al., 2025), which leverages a stronger VLM through teacher-student distillation. DHO can be viewed as a distillation-based framework with a post-hoc weighting strategy, whereas our method is a Bayesian-inspired, reliability-aware framework with sample-specific adaptive estimation.

As shown in Table 15, we compare four methods:

- CoOp based on RN50 CLIP, denoted as **CoOp (RN50)**.

- DHO with CoOp-RN50 as the student model and ViT-B/16 CLIP as the teacher, denoted as **DHO (S:CoOp-RN50,T:ViT-B/16)**.

- DHO+Ours, which integrates our Evidence-Prompt into DHO. In this setting, CoOp-RN50 is used as the student model, ViT-B/16 CLIP is used as the teacher, and the same ViT-B/16 CLIP also serves as the VLM prior for joint reliability estimation with noisy labels, denoted as **DHO+Ours (S:CoOp-RN50,T:ViT-B/16, P:ViT-B/16)**.

- Ours with CoOp-RN50 as the training model and ViT-B/16 CLIP as the VLM prior, denoted as **Ours (CoOp-RN50,P:ViT-B/16)**.

The results show that DHO substantially outperforms CoOp, indicating that distilling knowledge can effectively introduce useful prior information under noisy supervision. Moreover, DHO+Ours brings consistent additional gains over DHO. Meanwhile, Ours (CoOp-RN50,P:ViT-B/16) remains slightly better than DHO+Ours. One possible reason is that DHO transfers the VLM prior through distillation without explicitly assessing its sample-wise reliability, which may propagate potential bias from the prior during training. In contrast, Evidence-Prompt directly estimates the credibility of both the prior and noisy labels, leading to more reliable prior integration under noisy supervision.

*Table 15.* Comparisons of DHO and Ours method on **Caltech101** under Sym noise. Best results are highlighted in **bold**.

| METHOD | SYM | | | | | |
|---|---|---|---|---|---|---|
| | 12.5% | 25% | 37.5% | 50% | 62.5% | 75% |
| COOP(RN50) | 86.4 | 81.0 | 76.7 | 70.9 | 61.3 | 46.9 |
| DHO(S:COOP-RN50,T:VIT-B/16) | 95.0 | 92.9 | 93.0 | 92.2 | 90.2 | 88.7 |
| DHO+OURS(S:COOP-RN50,T:VIT-B/16,P:VIT-B/16) | 95.2 | 94.9 | 94.9 | 94.8 | 94.8 | 94.4 |
| OURS(COOP-RN50,P:VIT-B/16) | **95.3** | **95.6** | **95.7** | **95.1** | **95.2** | **94.9** |

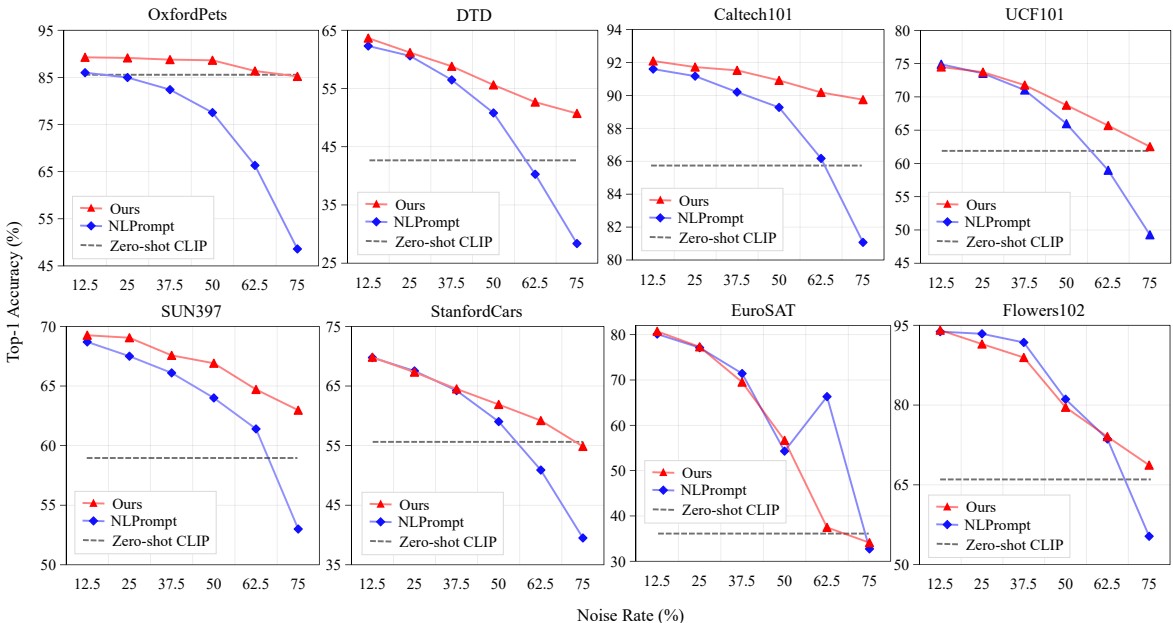

(a) Performance comparison of Zero-shot CLIP (Prior), NLPrompt, and our Evidence-Prompt on 8 datasets under Asym noise.

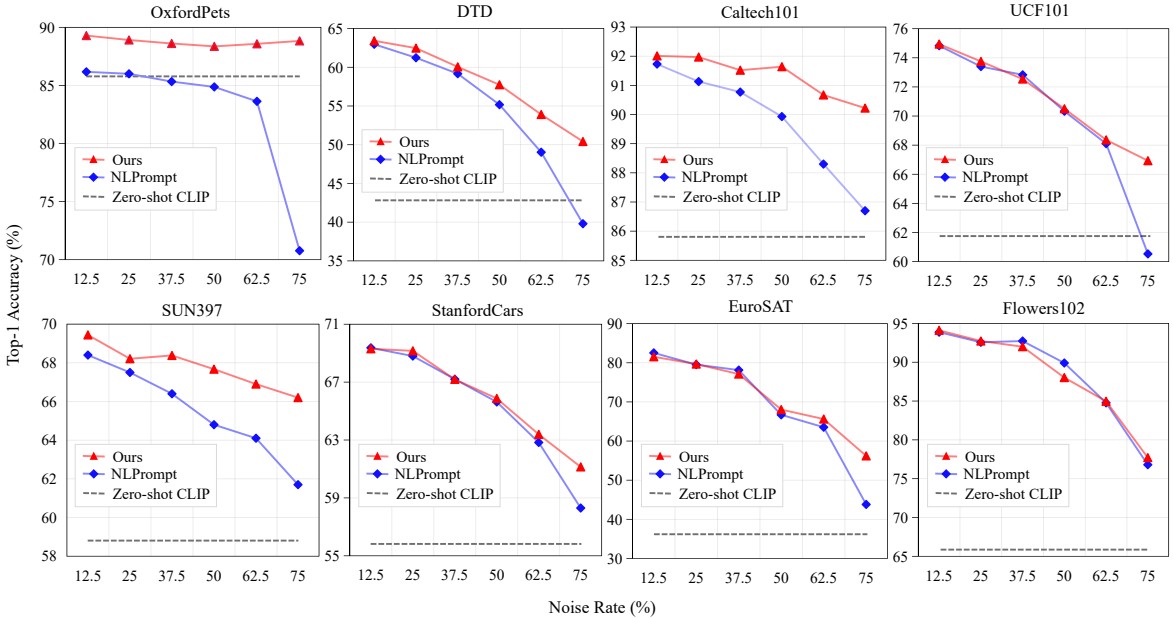

(b) Performance comparison of Zero-shot CLIP (Prior), NLPrompt, and our Evidence-Prompt on 8 datasets under Sym noise.

*Figure 6.* Performance comparison of Zero-shot CLIP (prior), NLPrompt (SOTA), and Evidence-Prompt under Sym/Asym noise across datasets and noise ratios.

