# OpenReview forum: "Leveraging Evidence Priors for Robust Prompt Learning under Noisy Supervision in Vision-Language Models"
_ICML.cc/2026/Conference — ICML 2026 regular_

### Official Review · Reviewer_7nZD · 2026-03-03

**Soundness:** 3
**Presentation:** 3
**Significance:** 3
**Originality:** 2
**Overall Recommendation:** 4
**Confidence:** 3

**Summary:**

This paper proposes Evidence-Prompt, a framework for robust prompt learning in VLMs under noisy labels.
The authors observe that frozen CLIP embeddings are resilient to label noise and can anchor learning.
Then authors propose sample-wise evidence priors from pre-trained representations, then performs credibility-aware reasoning that adaptively balances the prior and the noisy label via two credibility scores (supervision-agnostic and supervision-conditioned).
Experiments on 9 benchmarks including noisy-label dataset (Food101N) and relatively large-scale dataset (SUN397 in Appendix. B) under symmetric/asymmetric noise show promising results.

**Compliance With Llm Reviewing Policy:**

Affirmed.

**Final Justification:**

After the rebuttal, my concerns were resolved successfully.
So I raised the score, but reduced the confidence according to the second rebuttal comment.

**Key Questions For Authors:**

1.
How does Evidence-Prompt perform with other architecture such as ViT-B/16?
This would address if the proposed method generalizes well with different architecture family.

2.
Could you elaborate a little bit more of effect of each component in Table 5?
For example, removing E_je or E_sem under Asym-75% causes drops to ~10%.
Is this specifically because asymmetric noise creates consistent successor-class flips that fool the noisy label evidence?
A clearer explanation would help readers understand when each component is essential.

3.
Could Evidence-Prompt also effective in clean-label setting?
The complementary nature of foundation model priors and task-specific data itself was already discussed in the prior work [1], I wonder how Evidence-Prompt might work differently.

4.
Could you also provide a direct comparison with DHO [1] combined with Evidence-Prompt, if possible?
DHO also provides a framework to adaptively adjust the importance of the foundation model and the (possibly noisy) task-specific model at test time.
With that being said, I think additional discussion or experiment would benefit the contribution of this paper.

**Limitations:**

1.
Only single CNN-based backbone was evaluated (i.e., Resnet50).
I wonder this framework generalizes by doing analysis on different architecture such as ViT-based CLIP.

2.
The complementary nature of foundation model priors and task-specific data was already discussed in the prior work [1].
I think the paper might look like that it might be rather incremental work by setting the problem in noisy-label setting, though authors claim that such design was suggested for the first time in this context.

---

I'd be willing to discuss further during the rebuttal phase and would consider increasing the score if the above concerns are resolved well.

**Strengths And Weaknesses:**

## Strengths

-
The motivational claim is well-supported for necessity of pre-trained knowledge of VLM.
The adaptive weighting mechanism that assigns sample-wise credibility to the prior vs. the noisy label is well-motivated.

-
Analysis in the ablation (Table 5) shows the complementary roles of the four evidence terms across noise regimes.
The noisy-gradient suppression analysis (Fig. 3) further supports that the method successfully leverages the evidence prior.

-
The evaluation covers a good variety of datasets (9 total), including a real-world noisy-label dataset (Food101N) and a relatively large-scale dataset (SUN397 in the Appendix B), under both symmetric and asymmetric noise settings.

## Weaknesses

-
Only ResNet-50 CLIP is tested.
ViT-based CLIP is now the standard backbone and should be evaluated to confirm the approach generalizes.

-
I suspect the improvement is not limited to noisy settings.
Combining a general foundation model prior with task-specific data has a complementary nature that should also benefit clean datasets, as studied in previous work [1].
Thus, I believe proper justification and discussion would strengthen the contribution of the paper.

---

[1] Kang, Seongjae, et al. "Simple yet effective semi-supervised knowledge distillation from vision-language models via dual-head optimization." arXiv preprint arXiv:2505.07675 (2025).

---

> ### Author Rebuttal · Authors · 2026-03-31
>
> Thank you for your thoughtful feedback.
> ## **Q1 Comparisons based on the ViT-B/16**
> We further supplement comparisons using the **ViT-B/16** backbone, including CoOp, NLPrompt, and Ours, under the **SYM** noise setting.
>
> The results show that our method achieves the best performance on all datasets across all noise levels, demonstrating its strong generalization from RN50 CLIP to ViT-B/16.
>
> ***Dataset: OxfordPets***
>
> |Methods|12.5%|25%|37.5%|50%|62.5%|75%|
> |:-:|:-:|:-:|:-:|:-:|:-:|:-:|
> |CoOp|86.1|76.4|67.3|60.6|42.6|30.2|
> |NLPrompt|92.0|91.4|90.1|90.0|87.9|78.2|
> |**Ours**|**93.0**|**92.4**|**92.8**|**92.5**|**92.2**|**92.1**|
>
> ***Dataset: Caltech101***
> |Methods|12.5%|25%|37.5%|50%|62.5%|75%|
> |-|:-:|:-:|:-:|:-:|:-:|:-:|
> |CoOp|91.0|88.1|81.9|78.8|68.8|49.5|
> |NLPrompt|95.0|95.5|95.2|94.0|94.0|90.8|
> |**Ours**|**95.8**|**95.9**|**95.9**|**95.9**|**95.9**|**95.5**|
>
> ***Dataset: Food101N*** (a real-world noisy dataset)
> |CoOp|NLPrompt|**Ours**|
> |-|:-:|:-:|
> |78.2|81.9|**86.1**|
>  ## **Q2 The effect of each component**
>  From Table 5 in the main paper, we analyze the results from the following three perspectives.
>
> **1) Under Asym-75, $E_T^{\mathrm{je}}$ and $E_L^{\mathrm{sem}}$ are critical.**
> In this situation, incorrect labels often exhibit strong consistency, making external validation essential. $E_T^{\mathrm{je}}$ constrains the consistency between prior and support pool, while $E_L^{\mathrm{sem}}$ assesses whether the current label is supported by the prior semantic distribution. As a result, removing either term leads to a clear performance drop.
>
> **2) Under Sym noise, $E_T^{\mathrm{me}}$ plays a more prominent role.**
> Since Sym noise is random and incorrect labels lack stable structure, the prior's marginal confidence typically remains highly discriminative. Consequently, removing $E_T^{\mathrm{me}}$ leads to clear performance degradation under both Sym-50 and Sym-75, with a particularly pronounced drop at Sym-75.
>
> **3) Under Sym-50 and Asym-50, $E_L^{\mathrm{nl}}$ becomes more important.**
> Removing $E_L^{\mathrm{nl}}$ causes the largest drop under moderate noise, indicating that $E_L^{\mathrm{nl}}$ is informative in this situation. Moreover, the competitive performance of $E_L = E_L^{\mathrm{nl}}$ under low-noise settings suggests that $E_L^{\mathrm{nl}}$ captures most of the useful label-side evidence when label corruption is limited.
>
> |Dataset|Setting|Sym-12.5|Sym-25|Asym-12.5|Asym-25|
> |-|:-:|:-:|:-:|:-:|:-:|
> |EuroSAT|$E_L = E_L^{\mathrm{nl}}$|80.3|75.9|79.1|74.6|
> ||**Full**|**81.5**|**79.7**|**80.7**|**77.3**|
> |DTD|$E_L = E_L^{\mathrm{nl}}$|61.3|61.3|61.41|60.3|
> ||**FuLL**|**63.4**|**62.5**|**63.7**|**61.2**|
> |OxfordPets|$E_L = E_L^{\mathrm{nl}}$|88.4|88.0|88.8|88.0|
> ||**FuLL**|**89.3**|**88.9**|**89.3**|**89.1**|
>  ## **Q3 Comparisons under clean label setting**
>
> Our design is motivated by the noisy-label setting, where the key idea is to jointly model the reliability of both the prior distribution and the supervision signal. In the clean-label setting, label reliability is no longer needed. To enable a fair comparison, we replace label-reliability estimation with training-prediction reliability estimation, while retaining Evidence-Prompt for uncertainty-aware modeling. The results in the table below show that Evidence-Prompt remains effective, achieving the best performance on all datasets.
>
> |Methods|OxfordPets|DTD|Caltech101|Flowers102|StanfordCars|
> |-|:-:|:-:|:-:|:-:|:-:|
> |Zero-shot CLIP|85.8|42.7|85.8|66.0|55.8|
> |CoOp|88.7|63.7|91.8|94.5|72.9|
> |NLPrompt|86.6|63.4|91.0|94.5|70.1|
> |**Ours**|**89.9**|**65.2**|**92.7**|**95.0**|**73.6**|
>  ## **Q4 Comparisons on DHO Combined with Evidence-Prompt**
> We added the results for DHO and DHO+Ours under the noisy-label setting in the table below, with all datasets evaluated under Sym noise.
>
> **Method.** DHO leverages the VLM predictions to distill the current model, whereas our method explicitly models the reliability of both the VLM prior and the supervision signal.
>
> **Results.** Combining DHO with our method leads to further performance gains across different noise settings.
>
> **Implications.** In the noisy-label setting, DHO distills the model using the VLM prior, leveraging the VLM's label-noise-agnostic knowledge to improve robustness, while still being guided by noisy supervision. Our method, inspired by Bayesian belief updating, explicitly models the reliability of both the VLM prior and the supervision signal, making it complementary to DHO and further enhancing robustness against noisy labels.
>
> ***Dataset: DTD***
> |Methods|12.5%|25%|37.5%|50%|62.5%|75%|
> |-|:-:|:-:|:-:|:-:|:-:|:-:|
> |DHO|56.5|56.7|51.7|48.6|45.3|43.0|
> |**DHO+Ours**|**61.9**|**60.3**|**57.8**|**55.0**|**51.5**|**48.2**|
>
> ***Dataset: Caltech101***
> |Methods|12.5%|25%|37.5%|50%|62.5%|75%|
> |-|:-:|:-:|:-:|:-:|:-:|:-:|
> |DHO|91.6|89.5|88.4|87.0|86.5|86.0|
> |**DHO+Ours**|**92.4**|**90.8**|**89.5**|**88.8**|**88.1**|**87.8**|

---

> > ### Author Rebuttal · Reviewer_7nZD · 2026-04-02
> >
> > Thank you for the time and effort on the rebuttal with additional experiments to support the contribution.
> > And I appreciate that most of my concerns are resolved.
> >
> > But I have few concerns remained:
> >
> > (1; about Q3) Could you elaborate more on the training setting of clean-label setting experiment? For example, I don't get how training-prediction reliability is computed, and additional rationale about the effectiveness on the clean label setting.
> >
> > (2-1; about Q4) The noisy label setting seems to be the one that makes different from the prior work, but still VLM distillation and adaptive weighting (post-hoc) [1] between noisy-but-task-specific and noise-label-agnostic signals can mitigate such label noisy problem. could you elaborate more on the clear distinction about this?
> >
> > (2-2; about Q4) Also it seems the performance of DHO+ours seems to be worse than the performance reported in the main paper (Table 2), which is concerning to me. Additional rationale about this would be necessary to clearly distinguish the complementariness of this behavior.
> >
> > Thank you.

---

> > > ### Author Response · Authors · 2026-04-03
> > >
> > > We appreciate your careful review of our paper.
> > > ## **(1; about Q3)**:
> > > **Training Procedure:**
> > > 1) Our training procedure still **follows the Evidence-Prompt pipeline** and requires computing the supervision-agnostic credibility $E_T$ and the supervision-conditioned credibility $E_L$.
> > > 2) During training, $E_T$ is obtained exactly in the same way as in Section 4.3.2, i.e., derived from the VLM prior. For $E_L$, under the clean-label setting, the labels are fully reliable and thus no longer require explicit credibility evaluation. Accordingly, the focus naturally shifts to evaluating the reliability of the training model’s predictions, which corresponds to the **training-prediction reliability** mentioned in Q3. Specifically, in **Eq. (9)**, $\tilde{y}_i$ and $\tilde{y}_j$ now denote the predictions of the model from the previous training iteration on the current sample and its support-pool samples, respectively, from which $E_L^{\mathrm{nl}}$ is computed. Combined with $E_L^{\mathrm{sem}}$, it yields the supervision-conditioned credibility $E_L$.
> > >
> > > $E_{L,i}^{\mathrm{nl}}=\sum_{j\in\mathcal{S}(i)} w_{ij}\{I}(\tilde{y}_j=\tilde{y}_i) \quad (9)$
> > >
> > > 3) Then, in **Eq. (12)**, $y_s$ denotes the final output distribution obtained by combining model predictions $\tilde{y}_i$ with VLM prior $p^t_i$. Since labels are clean, the training loss is directly imposed between this output distribution and the **ground-truth labels**, to guide model predictions.
> > >
> > > $y_s = (1-e_i)\tilde{y}_i + e_i p^t_i \quad (12)$
> > > ## **(2; about Q4)**:
> > > Our noisy-label setting follows NLPrompt and related works. To the best of our knowledge, DHO does not directly evaluate under noisy-label settings, as it does not explicitly question the correctness of the given labels.
> > >
> > > The **key differences** are as follows.
> > >
> > > 1) Our method is a **reliability-aware uncertainty modeling framework**, whereas DHO is a **distillation-based framework**.
> > > 2) **DHO** uses the **VLM prior** as a **teacher** to distill the current model predictions.
> > > In contrast, **our method** treats the VLM prior and noisy labels as **two distinct yet potentially unreliable information sources**, viewing the former as the prior belief and the latter as the current observation, rather than using the prior as a teacher signal for distillation. Then, in our understanding, the "noisy-but-task-specific" and "noise-label-agnostic signals" in **DHO** correspond to the **predictions** from the **Supervised Head** and the **KD Head**, and are combined during inference via a **post-hoc weighting**.
> > > 3) In addition, whereas the post-hoc weighting $\alpha$ in DHO is **globally shared** across all samples, ours is a **sample-specific reliability estimate** that is **adaptively inferred** under our Bayesian-inspired framework.
> > >
> > > Therefore, DHO is a distillation-based framework with a post-hoc weight, whereas our method is a Bayesian-inspired, reliability-aware framework with sample-specific adaptive estimation.
> > > ## **(3; about Q4)**:
> > > **About Q4 experiments:** In Q4, the experiments for DHO and DHO+Ours are conducted using RN18, following the original DHO backbone while adopting our noisy-label dataloader, whereas Table 2 of the main paper uses CoOp based on RN50 CLIP as the backbone.
> > >
> > > **About complementary:** By "complementary", we mean that integrating our method into DHO can bring further gains over DHO alone. We apologize that our previous wording may have caused ambiguity.
> > >
> > > **Supplementary CoOp-backbone experiments:**
> > > To avoid ambiguity and further strengthen the comparison, we additionally conduct experiments under the **noisy-label setting** using **CoOp-RN50** as the backbone.
> > >
> > > We compare four methods:
> > >
> > > * **CoOp** based on RN50 CLIP, denoted as **CoOp (RN50)**;
> > > * **DHO** with CoOp-RN50 as the student model and ViT-B/16 CLIP as the teacher, denoted as **DHO (S:CoOp-RN50,T:ViT-B/16)**;
> > > * **DHO+Ours**: **DHO** with CoOp-RN50 as the student model and ViT-B/16 CLIP as the teacher, further integrated with our Evidence-Prompt, where ViT-B/16 CLIP serves as the VLM prior and is jointly assessed with noisy labels for reliability estimation, denoted as **DHO+Ours (S:CoOp-RN50,T:ViT-B/16, P:ViT-B/16)**;
> > > * **Ours** with CoOp-RN50 as training model and ViT-B/16 CLIP as the VLM prior, denoted as **Ours (CoOp-RN50,P:ViT-B/16)**.
> > >
> > > As shown, DHO outperforms CoOp. Furthermore, DHO+Ours brings clear additional gains, while Ours (CoOp-RN50,P:ViT-B/16) remains slightly better than DHO+Ours. One possible reason is that DHO relies on distilling VLM prior without assessing its reliability, which may introduce bias inherent to the prior during training.
> > >
> > > ***Dataset Caltech101*** on SYM noise
> > > |Method|12.5%|25%|37.5%|50%|62.5%|75%|
> > > |-|:-:|:-:|:-:|:-:|:-:|:-:|
> > > |CoOp(RN50)|86.4|81.0|76.7|70.9|61.3|46.9|
> > > |DHO(S:CoOp-RN50,T:ViT-B/16)|95.0|92.9|93.0|92.2|90.2|88.7|
> > > |DHO+Ours(S:CoOp-RN50,T:ViT-B/16,P:ViT-B/16)|95.2|94.9|94.9|94.8|94.8|94.4|
> > > |**Ours(CoOp-RN50,P:ViT-B/16)**|**95.3**|**95.6**|**95.7**|**95.1**|**95.2**|**94.9**|

---

### Official Review · Reviewer_vMmL · 2026-03-04

**Soundness:** 2
**Presentation:** 2
**Significance:** 3
**Originality:** 3
**Overall Recommendation:** 4
**Confidence:** 2

**Summary:**

The authors observe that frozen CLIP embeddings remain robust to label noise. They propose Evidence-Prompt that combines noise-agnostic prior and supervised signals. The method constructs soft training targets through a weighted interpolation between the CLIP prior and the noisy labels. Extensive experiments on eight benchmarks demonstrate significant improvements over existing methods.

**Compliance With Llm Reviewing Policy:**

Affirmed.

**Final Justification:**

After the second rebuttal, this work is boardline, so I raise the score, but reduce my confidence.

**Key Questions For Authors:**

See weaknesses.

**Limitations:**

The paper includes an Impact Statement, but it lacks a Limitations section.

**Strengths And Weaknesses:**

Strengths:

1, The observation that pretrained CLIP embeddings remain resilient to label noise makes sense. The prior yields performance gains.

2, The evidence prior module is plug-in, and it does not require any learnable components beyond the original prompt.

3, The method exhibits superior performance and robustness under high noise conditions.

Weaknesses:

1, The author overclaim the paper as a Bayesian reasoning framework, the core mechanism is a linear interpolation. And the authors acknowledge is not a strict Bayesian realization.

2, The writing of the manuscript, especially the introduction, needs improvement. For example, the description about Table 1 needs to summarize and generalize the insight, rather than simply listing the result values.

3, All experiments use only ResNet-50 as the CLIP image encoder. ViT-based (ViT-B/16, ViT-L/14) CLIP models are more common, but authors lack of them.

4, The method is reasonable but over engineering.

5, On the surface, there are not many hyperparameters, but at least 5 functions include implicit heuristic hyperparameters. These implicit hyperparameters lack both theoretical motivation and ablations.

---

> ### Author Rebuttal · Authors · 2026-03-31
>
> Thank you for your thoughtful feedback.
>  ## **Q1 On the Bayesian Claim**
>
> Strictly speaking, our method is not a rigorous realization of Bayesian inference. What we mean by 'Bayesian reasoning' is mainly a **Bayesian-inspired perspective**:
> * The pretrained prior is viewed as a **prior belief**, the noisy supervision is treated as an **observation**;
> * Their influence on the final target is adjusted through sample-wise **credibility estimation**.
>
> Thus, we draw from the core intuition of **Bayesian belief updating**, rather than adhering to a strictly defined probabilistic model.
>
> We will revise the wording in the paper to clarify this point.
>
>  ## **Q2 Introduction and Result Presentation**
> Thank you for this constructive suggestion. We will revise the relevant parts of the paper, especially the discussion of Table 1 in the main text, to more clearly highlight its **key takeaway**:
>
> * 1.The pretrained VLM prior provides a relatively stable semantic reference under noisy supervision.
> * 2.Naively introducing the prior yields modest gains and cannot account for reliability variations under unknown noise conditions.
> * 3.Inspired by a Bayesian perspective, we model the credibility of both information sources to enhance robustness under unknown noise.
>
>  ## **Q3 Comparisons based on the ViT-B/16**
>   We further supplement comparisons using the **ViT-B/16** backbone, including CoOp, NLPrompt, and Ours, under the **SYM** noise setting.
>
> The results show that our method achieves the best performance on all datasets across all noise levels, demonstrating its strong generalization from RN50 CLIP to ViT-B/16.
>
> ***Dataset: OxfordPets***
>
> |Methods|12.5%|25%|37.5%|50%|62.5%|75%|
> |-|:-:|:-:|:-:|:-:|:-:|:-:|
> |CoOp|86.1|76.4|67.3|60.6|42.6|30.2|
> |NLPrompt|92.0|91.4|90.1|90.0|87.9|78.2|
> |**Ours**|**93.0**|**92.4**|**92.8**|**92.5**|**92.2**|**92.1**|
>
> ***Dataset: Caltech101***
> |Methods|12.5%|25%|37.5%|50%|62.5%|75%|
> |-|:-:|:-:|:--:|:-:|:-:|:-:|
> |CoOp|91.0|88.1|81.9|78.8|68.8|49.5|
> |NLPrompt|95.0|95.5|95.2|94.0|94.0|90.8|
> |**Ours**|**95.8**|**95.9**|**95.9**|**95.9**|**95.9**|**95.5**|
>
> ***Dataset: Food101N***
> |CoOp|NLPrompt|**Ours**|
> |:-:|:-:|:-:|
> |78.2|81.9|**86.1**|
>  ## **Q4-Q5 Concern about Over-Engineering and Hyperparameters**
>
> Thank you for raising this concern. We would like to clarify that the performance gains of our method do not arise from complex engineering or extensive hyperparameter tuning.
> * **In fact, the only additional hyperparameter that needs to be manually specified is top-K.**
> * The learning rate $\lambda$ and the parameter $z$ in the SGCE loss are not specific to our method, but are standard settings. $z$ is the typical parameter required by all methods based on the GCE loss.
>
> * The $w$ in Eqs.(4) and (9), $\gamma$ in Eq.(10), and $e$ in Eq.(12) are **adaptive variables rather than tunable hyperparameters**. Specifically, $w$ is computed by Eq. (5), $\gamma$ is adaptively estimated by $E_L^{\mathrm{nl}}$ and $E_L^{\mathrm{sem}}$, and $e$ is predicted from $E_T$ and $E_L$ via Eq. (11). We acknowledge that the current notation may cause confusion. We will clarify this point in the revised manuscript.
>
> Moreover, our method consistently demonstrates stable improvements across different datasets, noise types, and noise levels. This suggests that the observed gains stem from the inherent effectiveness and robustness of the method, rather than from extensive tuning or dataset-specific engineering.

---

> > ### Author Rebuttal · Reviewer_vMmL · 2026-04-01
> >
> > Most of my other concerns have been addressed.
> >
> > However, evaluating the method only on ViT-B/16 is not sufficient. At least, the paper should include results on ViT-L/14 to demonstrate that the method remains effective beyond the base-scale backbone.
> >
> > If this is added and reported, I will increase my score.

---

> > > ### Author Response · Authors · 2026-04-02
> > >
> > > Thank you for your encouraging feedback. We are pleased to know that most of your concerns have been addressed.
> > >
> > > We also appreciate your suggestion to evaluate a larger-scale backbone.
> > > We have conducted additional **ViT-L/14** experiments on **OxfordPets and Caltech101** under **SYM noise from 12.5% to 75%**, as well as on the real-world noisy dataset **Food101N**. The results further demonstrate that our method remains effective beyond the base-scale backbone.
> > >
> > >
> > > ***Dataset: OxfordPets***
> > > |Methods|12.5%|25%|37.5%|50%|62.5%|75%|
> > > |-|:-:|:-:|:-:|:-:|:-:|:-:|
> > > |CoOp|84.8|78.4|69.6|61.1|47.1|30.9|
> > > |NLPrompt|93.8|92.9|93.2|92.0|91.2|83.7|
> > > |**Ours**|**95.3**|**94.9**|**94.9**|**94.6**|**94.5**|**94.1**|
> > >
> > >
> > > ***Dataset: Caltech101***
> > > | Methods  | 12.5% | 25%  | 37.5% | 50%  | 62.5% | 75%  |
> > > |-|:-:|:-:|:-:|:-:|:-:|:-:|
> > > | CoOp     |  91.2 | 85.8 |  80.7 | 74.7 |  61.4 | 42.5 |
> > > | NLPrompt |  97.2 | 97.4 |  96.8 | 95.2 |  94.8 | 94.5 |
> > > | **Ours**     |  **97.8** |**97.7** |  **97.7** | **97.5** |  **97.6** | **97.2** |
> > >
> > >
> > > ***Dataset: Food101N***
> > > | CoOp | NLPrompt | Ours |
> > > |-|:--------:|:----:|
> > > | 82.7 |   87.4   | **91.3** |

---

### Official Review · Reviewer_i2nS · 2026-03-07

**Soundness:** 3
**Presentation:** 3
**Significance:** 3
**Originality:** 3
**Overall Recommendation:** 4
**Confidence:** 4

**Summary:**

This paper studies prompt learning of vision-language models for noisy target datasets. Based on the observation that the pre-trained embeddings are resilient to label noise, this paper proposes Evidence-Prompt, a framework that enhances prompt learning by integrating stable pre-trained knowledge. Specifically, this work treats prompt learning as a Bayesian reasoning task, where credibility is derived from both supervision-agnostic and supervision-conditioned evidence. Experiments over multiple benchmarks show the effectiveness of the proposed method.

**Compliance With Llm Reviewing Policy:**

Affirmed.

**Final Justification:**

The authors' responses fully address my questions, so I will keep my positive score.

**Key Questions For Authors:**

Please see the weakness.

**Limitations:**

No. The paper only provides the Impact Statement.

**Strengths And Weaknesses:**

**Strength**

Extensive experiments under different noise setup over different benchmarks are provided, demonstrating the effectiveness of the proposed method.

The paper is well-structured and easy to follow.

**Weakness**

The main idea is to use VLM pre-trained knowledge to denoise by modulating predictions from noisy labels. This assumes that the VLM predictions are robust on the target domain. The paper should better quantify how much the VLM itself contributes. In particular, it would be important to report a simple baseline, that is, directly using the VLM to predict on the target dataset. If this baseline is already strong, it may suggest that noisy labels are not necessary. The paper should discuss this point more clearly.

The experiments include Symmetric Noise and Asymmetric Noise, but both are simulated noise settings. While simulations are useful for controlled analysis, results on real noisy datasets would be more convincing and should be provided if possible.

The paper mainly compares to prompt learning related methods. Since the paper targets denoising under label noise, it should also compare against classic learning-from-noisy-label methods. Even if most prior methods are not designed for VLMs, many denoising ideas could be adapted to the VLM setting.

---

> ### Author Rebuttal · Authors · 2026-03-31
>
> Thank you for your thoughtful feedback.
>
>  ## **Q1 Quantifying the Contribution of the VLM**
>
> To explicitly quantify the contribution of the VLM itself, we include **four groups of results** in the table below:
> (1) **zero-shot CLIP** results;
> (2) **CoOp** and **NLPrompt** results;
> (3) **CoOp + Naive prior** and **NLPrompt + Naive prior**;
> (4) **CoOp + Ours** and **NLPrompt + Ours**.
>
> These results assess the VLM’s contribution from two perspectives:
> (i) its **direct predictive ability**, reflected by zero-shot CLIP accuracy; and
> (ii) its **value as a prior**, reflected by the improvement brought by incorporating the VLM prior into prompt learning.
>
> From these results, we draw three main observations:
>
> * **The VLM prior provides a relatively stable reference for noisy-label learning.**
> CoOp + Naive Prior consistently outperforms CoOp across all noise ratios, while NLPrompt + Naive Prior mainly improves performance under high-noise conditions. This indicates that the VLM prior effectively mitigates performance degradation caused by noisy labels, particularly when supervision is heavily corrupted.
>
>
> * **However, directly using the VLM prior is not sufficient.**
>   Although Naive Prior is beneficial, its gains are limited and may even degrade performance under low-noise, especially for NLPrompt.
>
> * **Our method makes better use of the VLM prior and achieves the best overall performance.**
> By dynamically integrating the pretrained prior with noisy supervision through evidence-based credibility modeling, our method achieves more stable improvements across different noise levels and consistently outperforms when combined with either CoOp or NLPrompt.
>
> **Necessity of Noisy-Label Supervision.**
> These results suggest that the VLM prior is indeed useful, but **it cannot replace noisy label learning**. In practical scenarios, **the true noisy level of the training data is unknown beforehand**, making the VLM prior alone an unreliable source of supervision.
>
>
>  ***Dataset:DTD***
> **Zero-shot CLIP result: 42.7**
>
> |Method|Variant|12.5%|25%|37.5|50%|62.5|75%|
> |-|:-:|:-:|:-:|:-:|:-:|:-:|:-:|
> |CoOp|Original|56.0|49.6|43.3|34.4|27.8|17.3|
> ||+ Naive Prior|62.2|58.5|53.1|47.0|40.1|28.9|
> ||**+ Ours**|**63.4**|**62.5**|**60.1**|**57.7**|**53.9**|**50.4**|
> |NLPrompt|Original|63.0|61.2|59.2|55.2|49.0|39.8|
> ||+ Naive Prior|60.6|60.6|58.0|55.6|50.2|43.8|
> ||**+ Ours**|**63.1**|**63.1**|**60.8**|**58.5**|**55.9**|**51.7**|
>
> ***Dataset:OxfordPets***
> **Zero-shot CLIP result: 85.8**
>
> |Method|Variant|12.5%|25%|37.5%|50%|62.5%|75%|
> |-|:-:|:-:|:-:|:-:|:-:|:-:|:-:|
> |CoOp|Original|76.5|66.7|60.3|47.0|35.8|24.6|
> ||+ Naive Prior|86.1|83.7|80.9|76.1|69.1|66.5|
> ||**+ Ours**|**89.3**|**88.9**|**88.6**|**88.4**|**88.6**|**88.8**|
> |NLPrompt|Original|86.2|86.0|85.3|84.9|83.6|70.8|
> ||+ Naive Prior|85.2|85.4|84.7|83.1|81.9|78.2|
> ||**+ Ours**|**89.6**|**89.3**|**87.9**|**88.3**|**87.7**|**86.4**|
>
> ***Dataset:Food101N***
> **Zero-shot CLIP result: 77.6**
>
> |CoOp|CoOp+Naive Prior|CoOp+Ours|
> |-|:-:|:-:|
> |69.5|74.4|**79.2**|
> |**NLPrompt**|**NLPrompt+Naive Prior**|**NLPrompt+Ours**|
> |76.5|77.9|**78.3**|
>
>  ## **Q2 About real-world noisy dataset**
>  Food101N is a real-world noisy dataset, and we provide the performance comparison in Table 3 of the main paper.
>  ## **Q3 Comparisons with classic noisy-label learning methods**
> We further compare with classic noisy-label learning methods, including GCE[1], JAL-CE[2], JAL-FL[2], and OGC-CE[3]. All methods are adapted to the same VLM framework and evaluated on OxfordPets, DTD, EuroSAT, and the real-world noisy dataset Food101N, as shown in the table below. Our method consistently outperforms all competitors across all settings.
>
> ***Dataset: OxfordPets***
> |Methods|sym-25%|sym-50%|asym-25%|asym-50%|
> |-|:-:|:-:|:-:|:-:|
> |GCE|84.6|79.2|83.0|68.1|
> |JAL-CE|83.5|79.2|83.9|60.8|
> |JAL-FL|84.3|78.8|83.7|63.4|
> |OGC-CE|87.2|84.9|86.6|81.9|
> |**Ours**|**88.9**|**88.4**|**89.1**|**88.6**|
>
> ***Dataset: DTD***
> |Methods|sym-25%|sym-50%|asym-25%|asym-50%|
> |-|:-:|:-:|:-:|:-:|
> |GCE|59.8|50.7|57.6|44.0|
> |JAL-CE|60.7|54.2|59.8|43.2|
> |JAL-FL|59.4|51.4|60.8|45.0|
> |OGC-CE|58.5|54.9|59.9|46.2|
> |**Ours**|**62.5**|**57.7**|**61.2**|**55.6**|
>
> ***Dataset: EuroSAT***
> |Methods|sym-25%|sym-50%|asym-25%|asym-50%|
> |-|:-:|:-:|:-:|:-:|
> |GCE|78.6|63.1|72.7|45.3|
> |JAL-CE|77.8|59.5|75.2|42.7|
> |JAL-FL|77.8|61.1|73.1|41.2|
> |OGC-CE|78.6|61.1|72.3|33.1|
> |**Ours**|**79.7**|**68.0**|**77.3**|**56.7**|
>
> ***Dataset: Food101N***
> |GCE|JAL-CE|JAL-FL|OGC-CE|Ours|
> |-|:-:|:-:|:-:|:-:|
> |71.3|74.3|74.7|76.9|**79.2**|
>
> [1] Generalized cross entropy loss for training deep neural networks with noisy labels, NeurIPS2018
>
> [2] Joint Asymmetric Loss for Learning with Noisy Labels, ICCV2025
>
> [3] Optimized Gradient Clipping for Noisy Label Learning, AAAI2025

---

> > ### Author Rebuttal · Reviewer_i2nS · 2026-04-01
> >
> > Thanks for providing new experiments. I will keep my positive score.

---

> > > ### Author Response · Authors · 2026-04-01
> > >
> > > Thank you for your positive response to our paper and rebuttal. We sincerely appreciate the time and effort you devoted to reviewing our paper. We are also truly grateful for your constructive feedback and helpful suggestions, which have substantially improved the paper.

---

### Official Review · Reviewer_Rj7N · 2026-03-13

**Soundness:** 2
**Presentation:** 3
**Significance:** 2
**Originality:** 2
**Overall Recommendation:** 4
**Confidence:** 5

**Summary:**

This paper focuses on prompt learning for VLMs under noisy supervision. While prompt tuning offers a parameter-efficient way to adapt pretrained models for downstream tasks, its effectiveness drops as label noise increases. To address this issue, this paper introduces Evidence-Prompt, an approach that enhances prompt learning by integrating stable, noisy supervision-agnostic, pretrained knowledge.
Specifically, noisy supervised learning is reformulated as a Bayesian posterior reasoning problem treating the prior and the noisy supervision signal as two separate sources of observation, estimating their credibility, and performing evidence-conditioned posterior reasoning. Experiments are conducted using multiple datasets at various noise levels to demonstrate the effectiveness of the proposed approach.

**Compliance With Llm Reviewing Policy:**

Affirmed.

**Final Justification:**

The rebuttal addressed my initial concerns and hence I increased my score to weak accept.

**Key Questions For Authors:**

See the weaknesses mentioned above, specifically thorough evaluation of naive prior approach and potential issues with the design of `E_{L,i}`

**Limitations:**

Yes

**Strengths And Weaknesses:**

**Strengths**
* The paper is written well and was easy to follow
* Experiments were conducted on multiple datasets under several label noise settings
* The proposed approach is shown to outperform various existing approaches.

**Weaknesses**

**Unconvincing experimental results:**
* Results for simple naive prior approach, i.e., giving equal weight to both clip prior and label supervision are reported only under the extreme noise cases (Tables 1 and 5) and only with COOP approach. In order to concretely demonstrate that the proposed approach is indeed useful to the community, authors should include the results for following strategies in Table 2 and Table 3: COOP + Naive Prior, JoAPR + Naive Prior and NLPrompt + Naive prior. NLPrompt seems to be the best among existing approaches. So, as a practitioner, I would at least first try NLPrompt + naive prior before testing more sophisticated approaches. Most of the ablations are done at 50% and 75% noise levels. These are extreme noise levels and do really reflect real world use cases. Results at 12-25% noise ratios are more reflective of usefulness of this work.

**Potential issues with the design of E_{L,i}:**
* Consider the scenario where the supervision for a sample and its neighbors is correct, i.e., the label `\tilda{y}_i` is correct and the labels of its neighbors are also correct, but the pretrained CLIP model is confidently wrong, i.e., high value of `E_{T,i}` but low `p_i^t(\tilda{y}_i)` and `q_i^t(\tilda{y}_i`) values. In this case, the value of `E_{L,i}^{sem}` will be low and the value of `E_{L,i}^{nl}` will be high. Ideally, since the supervision is correct in this case, we would want the supervision-conditioned credibility `E_{L,i}` to be high, but according to eq.(10), `E_{L,i}` will be close to `E_{L,i}^{sem}` which is low. This is happening because the computation of `E_{L,i}` is biased towards `E_{L,i}^{sem}` whenever `E_{L,i}^{sem}` and `E_{L,i}^{nl}` differ. Due to this, the computed posterior distribution `y_s` in Eq.12 is biased towards the wrong prior even when the supervision is correct and the pretrained CLIP model is wrong, which is undesirable.

* The pretrained model belief is already being captured in Eq (12) via the supervision-agnostic path. So, is there a need for using the prior model's belief (`p_i^t(\tilda{y}_i)` and `q_i^t(\tilda{y}_i)` values) again in computing `E_{L,i}`? This feels like double counting or giving too much weight to the pretrained model's prior. If pretrained model is already fairly good on the target task and noise level is too high, may be this double counting is reasonable. But, if the pretrained model is not great on the target task and we do not know the actual label noise level, then giving too much importance to prior may not be good. To understand things better, authors should (1) report the zero-shot accuracy of the CLIP model on the training data they have used in each experiment (2) do more experiments with `E_{L,i} = E_{L,i}^{nl}` (more datasets and lower noise settings beyond what is presented in Table 5)

**Noise details**
* How is asymmetric noise case implemented? The text simply says: samples from the same class are consistently mapped to a neighboring, similar class (the successor class). I can imagine two ways to do this: (1) for every class, select 75% samples and change their labels consistently, or (2) choose 3/4th classes and change the labels of all samples in these classes using the same consistency rule. Which one did the authors follow?

---

> ### Author Rebuttal · Authors · 2026-03-31
>
> Thank you for your thoughtful and encouraging feedback.
> ## **Q1 Unconvincing experimental result**
> ### **1) Experiments on NLPrompt, CoOp, and their + Naive Prior/+ Ours Variants**
> We report the comparison of these variants.
>
> **DTD** on SYM noise
> |Method|12.5%|25%|37.5|50%|62.5|75%|
> |-|:-:|:-:|:-:|:-:|:-:|:-:|
> |CoOp|56.0|49.6|43.3|34.4|27.8|17.3|
> |+ Naive Prior|62.2|58.5|53.1|47.0|40.1|28.9|
> |**+ Ours**|**63.4**|**62.5**|**60.1**|**57.7**|**53.9**|**50.4**|
> |NLPrompt|63.0|61.2|59.2|55.2|49.0|39.8|
> |+ Naive Prior|60.6|60.6|58.0|55.6|50.2|43.8|
> |**+ Ours**|**63.1**|**63.1**|**60.8**|**58.5**|**55.9**|**51.7**|
>
> **OxfordPets** on SYM noise
> |Method|12.5%|25%|37.5%|50%|62.5%|75%|
> |-|:-:|:-:|:-:|:-:|:-:|:-:|
> |CoOp|76.5|66.7|60.3|47.0|35.8|24.6|
> |+ Naive Prior|86.1|83.7|80.9|76.1|69.1|66.5|
> |**+Ours**|**89.3**|**88.9**|**88.6**|**88.4**|**88.6**|**88.8**|
> |NLPrompt|86.2|86.0|85.3|84.9|83.6|70.8|
> |+ Naive Prior|85.2|85.4|84.7|83.1|81.9|78.2|
> |**+Ours**|**89.6**|**89.3**|**87.9**|**88.3**|**87.7**|**86.4**|
>
> **Food101N**
> |CoOp|+Naive Prior|+Ours|
> |-|:-:|:-:|
> |69.5|74.4|**79.2**|
> |**NLPrompt**|**+Naive Prior**|**+Ours**|
> |76.5|77.9|**78.3**|
>
> From tables, **we draw two key observations**:
> * **Prior provides a relatively stable reference for noisy label learning**
>
> CoOp+Naive Prior improves consistently across all noise ratios. NLPrompt+Naive Prior mainly benefits high-noise.
> * **Evidence Prompt brings consistent gains**
>
> Across all CoOp and NLPrompt results, +Ours consistently outperforms +Naive Prior. For NLPrompt, +Naive Prior may hurt in low to moderate noise.
> ### **2) The Ablation Studies to Lower Noise Ratios.**
> Under **lower noise ratios**, removing $E_L^{\mathrm{nl}}$ causes the largest drop, showing that label-side information is more important in this regime. This is consistent with the intended roles of the evidence terms.
> |Setting|Sym-12.5|Sym-25|Asym-12.5|Asym-25|
> |-|:-:|:-:|:-:|:-:|
> |w/o $E_T^{\mathrm{me}}$|80.4|73.3|78.1|71.8|
> |w/o $E_T^{\mathrm{je}}$|80.8|75.0|78.9|73.0|
> |w/o $E_L^{\mathrm{nl}}$|73.7|68.7|71.7|67.2|
> |w/o $E_L^{\mathrm{sem}}$|80.3|75.9|79.1|74.6|
> |**Full**|**81.5**|**79.7**|**80.7**|**77.3**|
>
> ## **Q2 Potential issues with the design of $E_{L,i}$**
> ### **1) The Case Where the Prior and Labels Are in Complete Conflict**
> The reviewer raises a concern about an extreme case.
>
> First, our method uses the full class distribution instead of top-1, which mitigates the risk of extreme cases. As shown below, CLIP's Top-10 accuracy exceeds 90% on all datasets, suggesting that even top-1 is wrong, the prior still retains useful semantic information. Since our method is distributional rather than hard-prediction view, an incorrect top-1 may not undermine evidence construction or posterior computation.
> |Dataset|Top1|Top5|Top10|
> |-|:-:|:-:|:-:|
> |UCF101|61.8|84.9|91.6|
> |stanford_cars|55.8|89.6|95.5|
> |oxford_pets|85.8|95.1|97.1|
>
> Second, we examine the occurrence of this case on EuroSAT. Its ratio is 1.5% at 12.5 noise to 0% at 75 noise. Inspired by Bayesian belief updating, our method treats prior and supervision as two information sources under unknown noise, for more robust learning.
> |Noise Rate|Case Count|Total Count|Ratio(%)|
> |-|:-:|:-:|:-:|
> |12.5|487|32k|1.5|
> |50|4|32k|0.01|
> |75|0|32k|0.00|
> ### **2) Concern on Double Counting the Prior**
> **The use of the prior in Eq. (12) and $E_{L,i}$ is different**. In Eq. (12), the prior distribution $p_i^t$ constructs the posterior target. In contrast, $E_{L,i}$ uses $p_i^t(\tilde y_i)$ and $q_i^t(\tilde y_i)$ to evaluate the credibility of the noisy label based on semantic consistency, rather than reintroducing the prior as a target.
>
> Then, **under noisy-label settings, the reliability of the label is entirely unknown**.
> The prior as a semantic reference is relatively independent of current label environment, providing an additional semantic constraint not double counting.
>
> Following the reviewer’s suggestion, we report:
>
> **1) Zero-shot CLIP results on training data** is not good.
>
> |EuroSAT|OxfordPets|DTD|Caltech101|UCF101|Flowers102|StanfordCars|FooD101N|
> |:-:|:-:|:-:|:-:|:-:|:-:|:-:|:-:|
> |21.3|60.9|30.3|62.3|41.9|48.9|38.6|60.9|
>
> **2) More ablations with $E_{L,i}=E_{L,i}^{nl}$ on more datasets under lower noise.** Under low noise, $E_L^{\mathrm{nl}}$ is slightly worse than the full method, the additional gain of the full method reflects its stronger ability to adaptively integrate multiple evidence across all settings.
> |Dataset|Setting|Sym-12.5|Sym-25|Asym-12.5|Asym-25|
> |-|:-:|:-:|:-:|:-:|:-:|
> |EuroSAT|$E_L = E_L^{\mathrm{nl}}$|80.3|75.9|79.1|74.6|
> ||**Full**|**81.5**|**79.7**|**80.7**|**77.3**|
> |DTD|$E_L = E_L^{\mathrm{nl}}$|61.3|61.3|61.41|60.3|
> ||**Full**|**63.4**|**62.5**|**63.7**|**61.2**|
> |OxfordPets|$E_L = E_L^{\mathrm{nl}}$|88.4|88.0|88.8|88.0|
> ||**Full**|**89.3**|**88.9**|**89.3**|**89.1**|
>
> ## **Q3 Noise details**
> Asym noise setting follows NLPrompt and corresponds to the first case you mentioned. We will clarify this in the revised version.

---

> > ### Author Rebuttal · Reviewer_Rj7N · 2026-04-01
> >
> > I thank the authors for their rebuttal which answered most of my question. Hence, I am increasing my score.

---

> > > ### Author Response · Authors · 2026-04-01
> > >
> > > Thank you for your positive response to our rebuttal. We deeply appreciate the time and effort you have dedicated to reviewing our paper. We also greatly appreciate your constructive feedback and helpful suggestions, which have helped us improve the paper substantially.

---

### Decision · Program_Chairs · 2026-04-30

**Decision:**

Accept (regular)

**Comment:**

The paper tackles the problem of prompt learning under label noise. The paper receives four reviews with 4x weak accept ratings. Overall, the reviewers are positive about this work: they find the idea of exploiting prior knowledge in VLM to overcome label noise interesting and the experiments extensive. The main concern raised during the review period is that the results were not comprehensive enough. Specifically, important baselines such as those based on combining prompt learning with label-noise strategies were missing, and ViT-based backbones were not included in the experiments. The rebuttal has provided additional results as requested by the reviewers and the main concern was well resolved. Based on the positive reviews, the AC recommends acceptance. The AC would like to request that the additional results presented in the rebuttal be included in the final version.